# Left–Right Symmetry Breaking in CLIP-Style Vision-Language Models Trained on Synthetic Spatial-Relation Data

**Takaki Yamamoto** [1]  **Chihiro Noguchi** [1]  **Toshihiro Tanizawa** [1]

## Abstract

Spatial understanding remains a key challenge in vision-language models. Yet it is still unclear whether such understanding is truly acquired, and if so, through what mechanisms. We present a controllable 1D image–text testbed to probe how left–right relational understanding emerges in Transformer-based vision and text encoders trained with a CLIP-style contrastive objective. We train lightweight Transformer-based vision and text encoders end-to-end on paired descriptions of one- and two-object scenes and evaluate generalization to unseen object pairs while systematically varying label and layout diversity. We find that contrastive training learns left–right relations and that label diversity, more than layout diversity, is the primary driver of generalization in this setting. To gain the mechanistic understanding, we perform an attention decomposition and show that interactions between positional and token embeddings induce a horizontal attention gradient that breaks left–right symmetry in the encoders; ablating this contribution substantially reduces left–right discrimination. Our results provide a mechanistic insight of when and how CLIP-style models acquire relational competence.

## 1. Introduction

Vision-language models (VLMs) have recently achieved striking progress across a wide range of multimodal tasks (Radford et al., 2021; Li et al., 2022; Du et al., 2022; Wang et al., 2022; Ghosh et al., 2025; Sapkota et al., 2025). Yet a growing body of evidence indicates that these systems struggle with relational understanding—recognizing who is doing what to whom and how objects are arranged with respect to one another—as well as with geometric or

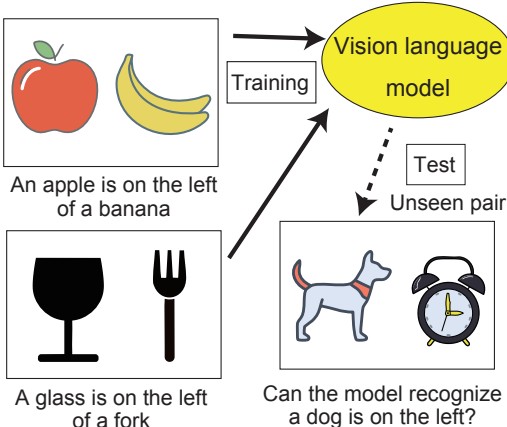

*Figure 1.* Schematic of our research question: how spatial and relational capabilities are, or are not, acquired in VLMs.

spatial reasoning more broadly (Yuksekgonul et al., 2023; Kamath et al., 2023; Huang et al., 2025; Stogiannidis et al., 2025; Cheng et al., 2024; Chen et al., 2025), and that these limitations are closely tied to persistent gaps in their compositional generalization (Thrush et al., 2022; Ma et al., 2023; Ray et al., 2023; Tong et al., 2024; Zheng et al., 2024). Beyond academic interest, safety-critical applications such as autonomous driving and robotic manipulation demand accurate spatial reasoning: inferring relative positions (left/right, in front of/behind), topological relations (inside/around/overlapping), and maintaining consistency across viewpoint and layout changes.

Large-scale VLMs trained on web-scale image–text corpora have motivated a wave of benchmarks targeting compositional and spatial reasoning (Johnson et al., 2017; Suhr et al., 2017; 2019; Thrush et al., 2022; Ma et al., 2023; Ray et al., 2023), which expose substantial performance gaps across architectures and training regimes. For instance, Yuksekgonul et al. (2023) introduce the Attribution, Relation, and Order (ARO) benchmark and demonstrate that state-of-the-art VLMs behave like "bag-of-words" models: they exhibit poor relational understanding, struggle to bind objects to their attributes, and show severe insensitivity to word order. The authors attribute this failure to the fact that standard retrieval objectives can be solved without leveraging com-

[1]InfoTech, Toyota Motor Corporation. Correspondence to: Takaki Yamamoto <takaki_yamamoto@mail.toyota.co.jp>.

*Proceedings of the 43rd International Conference on Machine Learning*, Seoul, South Korea. PMLR 306, 2026. Copyright 2026 by the author(s).

positional structure, removing the incentive for models to learn it. Controlled synthetic evaluations in CLEVR-style environments (Johnson et al., 2017), together with naturalistic benchmarks like NLVR2 (Suhr et al., 2019) and Winoground (Thrush et al., 2022), corroborate these findings: contrastive pretraining on captions provides limited explicit relational supervision.

Despite rapid empirical progress, we still lack a mechanistic account of how spatial and relational capabilities are— or are not—acquired in VLMs (Fig. 1). Interpretability work demonstrates the value of carefully designed toy tasks and small models for reverse-engineering emergent behaviors (Elhage et al., 2022; Okawa et al., 2023; Csordás et al., 2024; Rai et al., 2025), but controlled studies that target spatial cognition within the VLM pipeline remain sparse. Recent work has begun to disentangle the contributions of different components—for instance, Qi et al. (2025) show that vision token embeddings suppress positional information in the LLM, while others attribute spatial failures primarily to training data rather than architecture (Chen et al., 2024)—yet a unified understanding of where spatial signals arise, how they are transformed, and which components are necessary or sufficient remains elusive.

In this work, we begin with CLIP-style contrastive models (Radford et al., 2021) as one of the fundamental VLM architectures. While recent work has shown that downstream components such as causally-structured decoders can improve compositional performance even with a frozen CLIP visual encoder (Parascandolo et al., 2025), the question of how spatial understanding emerges in the encoder itself remains open. We take a bottom-up approach and ask: can CLIP-style Transformers learn faithful encodings of relative spatial relations, and by what mechanism? In a minimal synthetic 1D image–text setting, we train Transformer-based vision and text encoders end-to-end with a contrastive objective, evaluate generalization to unseen object pairs under varying label and layout diversity, and conduct mechanistic analyses. We also confirm that a similar mechanism appears in an autoregressive VLM setting (Sec. 9). Our main contributions are as follows:

- We present a controllable testbed for analyzing the emergence of left–right spatial understanding in CLIP-style Transformers, built on a synthetic 1D image–text dataset with one or two objects.

- We reveal that CLIP-style training learns left–right relations of objects in images and generalizes to unseen object pairs. By varying label diversity and layout diversity, we find that label diversity is the primary driver of generalization in our setup.

- We provide a mechanistic account: decomposing per-head pre-softmax attention logits reveals that interac-

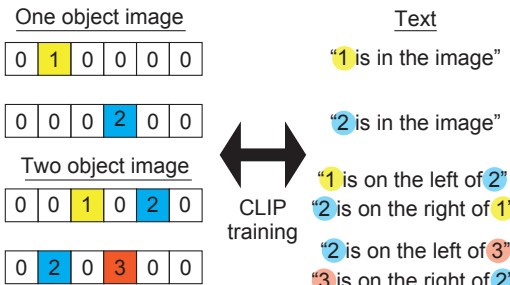

*Figure 2.* Schematic of CLIP model training of the toy dataset of 1D images and the corresponding texts. For two object images, *left* and *right textual representations* denote inverse relations that describe the same configuration.

tions between positional and token embeddings induce a horizontal attention gradient that breaks left–right symmetry; ablating this contribution reduces left–right discrimination.

## 2. Experimental Setup

We introduce our toy experimental setting, shown in Fig. 2, to study how CLIP-style Transformer-based VLM models learn spatial relations between objects in images. We evaluate a CLIP-style contrastive learning setup between one-dimensional images and text.

### 2.1. Synthetic 1D Image-Text Dataset

**Image** An image is a 1D sequence of length $D^{\text{image}}$. Each object occupies only one pixel. We consider two types of images: (i) single-object images, and (ii) two-object images in which each of the two distinct objects occupies one pixel. Background pixels take value 0, and objects are encoded as integers $\geq 1$, with different integers denoting different object categories. We consider $N_{\text{tot}}$ object categories, labeled 1 through $N_{\text{tot}}$. A learnable class token [CLS] is prepended at the first position of the image sequence. We fix $D^{\text{image}} = 10$.

**Text** For single-object images, the caption is "[label index] is in the image." and is formalized as the token sequence [label index] [is] [in the image] [EOT]. For two-object images, we define two caption formats. The *left textual representation* follows the template: [label index 1] [is on the left of] [label index 2] [EOT], while the *right textual representation* follows: [label index 2] [is on the right of] [label index 1] [EOT]. To simplify the problem, we regard [in the image] and [is on the left/right of] as single tokens. For brevity, we write XLY for "X is on the left of Y" and XRY for "X is on the right of Y."

**Training and Test Dataset** To learn single-object recognition, we generate $n_1$ images for labels from 1 to $N_{\text{tot}}$ by placing each object at random positions along the 1D

sequence. To learn spatial relations, we select a subset of object labels from 1 to $N_{\text{pair}}$ and generate $n_2$ images for every ordered pair from this subset. For each image, we randomly sample two distinct positions $p_1 < p_2$ and place the first object at $p_1$ and the second at $p_2$, pairing the image with the text "[label of the first object] [is on the left of] [label of the second object] [EOT]". We keep the remaining $N_{\text{val}} = N_{\text{tot}} - N_{\text{pair}}$ labels from $N_{\text{pair}} + 1$ to $N_{\text{tot}}$ for the test dataset to evaluate how much the trained model generalizes to the labal pairs unseen during the training. We fix $n_1 = 5$ and $N_{\text{val}} = 5$. A schematic overview is provided in App. A.

## 2.2. Model and Training

**Model** Both the vision and text encoders are Transformer-based (Vaswani et al., 2017). The vision encoder uses bidirectional self-attention (no causal mask), while the text encoder uses a causal mask. They consists of $M_{\text{rep}}$ repetitions of $M_B$ Transformer blocks, each comprising multi-head self-attention with $M_h$ heads, residual connections, multi-layer perceptron (MLP) with GeLU activation function. For multi-head attention, the dimension of embeddings is $d_{\text{head}}$ for each head. The hidden dimension of MLP is $d_{\text{MLP}}$. The dimensions of the token embeddings, positional embeddings and embeddings of attention layers are $d_{\text{model}}$. The token and positional embeddings are learnable both for vision and text encoders. LayerNorm and dropout with rate $p$ are also applied. The image representation is taken from the output at the class token [CLS], while the text representation is taken from the [EOT] token. When the vision and text encoders use different hyperparameters, we annotate them with subscripts "vis" and "txt"; otherwise, a single shared setting applies. For mechanistic analyses (Sections after Sec. 4), we adopt a simplified model without LayerNorm and MLP layers to reduce nonlinearity and isolate attention dynamics (Elhage et al., 2021).

**Training** We apply a linear projection to the text representation and compute the cosine similarity with the image representation. The model is trained with the standard CLIP contrastive loss over batchwise image–text similarities. The loss function is the average of the cross-entropy losses computed over image-to-text and text-to-image logits.

Unless otherwise noted, training hyper-parameters are: learning rate $lr = 1 \times 10^{-4}$ (AdamW), weight decay $w = 0.2$, number of epochs $N_{\text{epoch}} = 10,000$, and $p = 0.1$. We use batch size $b_s = 50$ for experiments with only-left textual representation, while we use $b_s = 100$ for those with left and right textual representations. See App. B for details.

## 2.3. Generalization

We evaluate the following three types of generalization by computing the cosine similarity between image and text rep-

resentations (Schematic overview in App. A). Specifically, we use image-to-text retrieval accuracy: for each image, we determine whether the correct text achieves the highest cosine similarity among all texts existing in the training and validation sets, including both texts for one-object and two-object images.

**Single-Object Positional Generalization**: We define "single-object positional generalization" as the model's ability to correctly match text descriptions for single-object images placed at positions unseen during training. We randomly sampled $n_{\text{val}} = 5$ images per label, disjoint from the training set, to form the validation dataset.

**Seen-Pair Configuration Generalization**: "Seen-pair configuration generalization" refers to whether the model correctly matches the corresponding text when given images of ordered label pairs seen during training but placed in novel relative configurations (i.e., unseen position pairs). To evaluate this, we constructed a validation set by randomly sampling $n_{\text{val}} = 5$ images per label pair at held-out configurations, disjoint from the training set.

**Unseen-Pair Generalization**: "Unseen-pair generalization" refers to whether the model correctly matches the corresponding text when given ordered label pairs not seen during training. Concretely, we hold out the remaining $N_{\text{val}}$ object categories from the two-object training set and construct a validation set by randomly sampling $n_{\text{val}} = 5$ images per ordered pair drawn from these held-out categories, with positions sampled as described above.

In experiments using both *left* and *right textual representations* in Sec. 7, each two-object image is associated with two semantically equivalent captions (See Fig. 2). Accordingly, we consider the image-text matching to be correct if both representations rank within the top-2 similarity scores.

## 3. Generalization in Recognition of Relative Position in CLIP-Style VLM

In this section, we investigate whether the three types of generalization defined in Sec. 2.3 emerge under a CLIP-style training setup. We here use only the *left textual representation* as the caption for two-object images, while we show the result with both *left* and *right textual representations* in Sec. 7. Figure 3 (left panels) shows image-to-text alignment accuracy as a function of the number of training labels, serving as a proxy for these generalization behaviors. The number of spatial configurations of objects is also varied. We observe that training the model on a larger variety of object categories enhances all three types of generalization and leads to high accuracy. In contrast, variation in the spatial arrangement of objects has little effect on performance.

Figure 3 (right panels) visualizes the cosine-similarity ma-

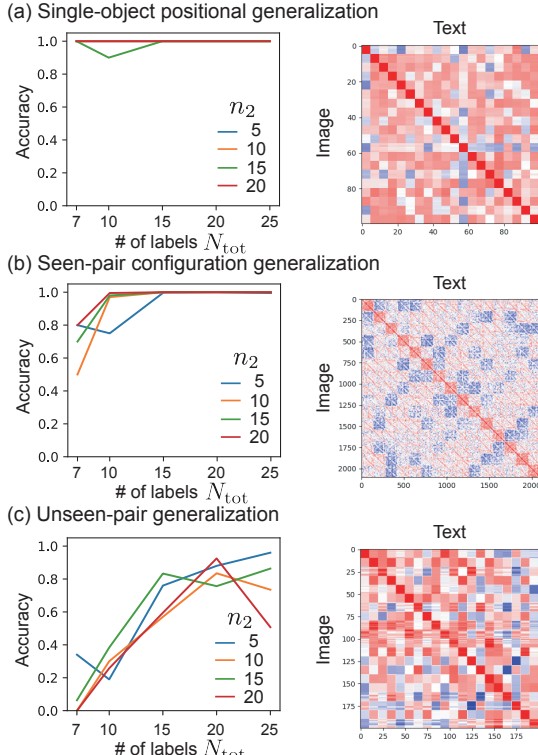

Figure 3. Three types of generalization observed in a CLIP-style training setup. This experiment is performed only with the *left textual representation*. (Left) Accuracy is shown for three types of generalization (a–c). (Right) Cosine similarity maps between image and text embeddings from the output layers are shown ($N_{\text{tot}} = 20, N_{\text{pair}} = 15, N_{\text{val}} = 5, n_2 = 10$). In the similarity map of (a), $n_1$ images sharing the same text representation are repeated $N_{\text{tot}}$ times along the image axis. In (b) and (c), $n_2$ images with the same text representation are repeated for all ordered pairs from $N_{\text{pair}}$ and $N_{\text{val}}$ labels, respectively. Hyperparameters: $M_B = 2, M_{\text{rep}} = 2, M_h = 4, d_{\text{head}} = 32, d_{\text{MLP}} = 512, d_{\text{model}} = 128$.

trix between image and text embeddings. Pronounced diagonal block patterns indicate correct image–text alignment. We found that weight decay regularization enhances the generalization (see App. C), suggesting that the model learns generalizable algorithms by avoiding overfitting, reminiscent of observations in the grokking literature (Power et al., 2022). Also, the analysis of the dynamics of generalization accuracy and the training loss reveals that single-object positional, two-object coordination, and unseen object pair generalization are achieved consecutively (App. D).

## 4. Left-Right Symmetry Breaking in Attention Patterns of Vision Encoder of Generalized Models

To probe what algorithm the model learns to give rise to these forms of generalization, we simplify the architec-

ture (Elhage et al., 2021). Specifically, we ablate components such as LayerNorm and MLP, reduce the network to a single Transformer block ($M_B = M_{\text{rep}} = 1$), and repeat the experiments. As shown in Fig. 4(a), despite this simplification, we observe a qualitatively similar behavior to that of the original model studied in Sec. 3. We show the generalization dynamics in App. D.

We probe the model's internal mechanism by visualizing attention patterns, computing the attention weight matrix $A^h = \text{Softmax}_{\text{row}}\left(Q^h K^{h\top}/\sqrt{d_{\text{head}}}\right)$ for the $h$-th head of the Transformer block. We hereafter drop the index $h$ of heads. The query and key matrices are computed through linear transformations of the input: $Q = XW_Q^T + B_Q^T, K = XW_K^T + B_K^T$, where $X \in \mathbb{R}^{n \times d_{\text{model}}}$ represents the sum of the token embeddings $E \in \mathbb{R}^{n \times d_{\text{model}}}$ of the input sequence and the learnable positional embedding $P \in \mathbb{R}^{n \times d_{\text{model}}}$. $n$ is the input length of the Transformer. $W_Q, W_K \in \mathbb{R}^{d_{\text{head}} \times d_{\text{model}}}$ are the learned query and key projection matrices. $B_Q, B_K \in \mathbb{R}^{d_{\text{head}} \times n}$ are the bias vectors for queries and keys, where the same bias vectors $b_Q$ and $b_K \in \mathbb{R}^{d_{\text{head}}}$ are applied to every token position in the sequence. The row-wise softmax operation ensures that each row of $A$ sums to 1, representing a probability distribution over which input tokens to attend to. The resulting attention matrix $A \in \mathbb{R}^{n \times n}$ contains the attention scores, where element $A_{ij}$ indicates how much the $i$-th token attends to the $j$-th token.

In Fig. 4(b), we show example attention patterns from the vision encoder of a trained model that generalizes well. The two example images both contain the label pair (17, 19), with their spatial positions swapped; notably, this pair was not included in the training set. Across all attention heads, we observe that the class token consistently attends to object locations (Recall that in a 1-layer Transformer, the final representation is derived solely from the class token's output.). Notably, head 2 exhibits strong attention toward the right-side object, regardless of whether it is label 17 or 19. This suggests that the head's function is relational rather than label-specific: it selectively attends to the right-side object independent of identity.

Figure 4(c) presents the left-right attention bias statistics for each of the four heads. The bias is quantified by determining, for each head, whether the left or right object receives maximal attention from the class token, and computing the proportion across test samples. Head 2 displays the strongest right-bias, while the remaining heads exhibit left-biased attention patterns. This functional specialization — with certain heads selectively attending to right-side objects and others to left-side objects — enables the model to encode the relative spatial positions of objects. Without such left-right symmetry breaking, the vision encoder would be unable to distinguish which object is on the left or right.

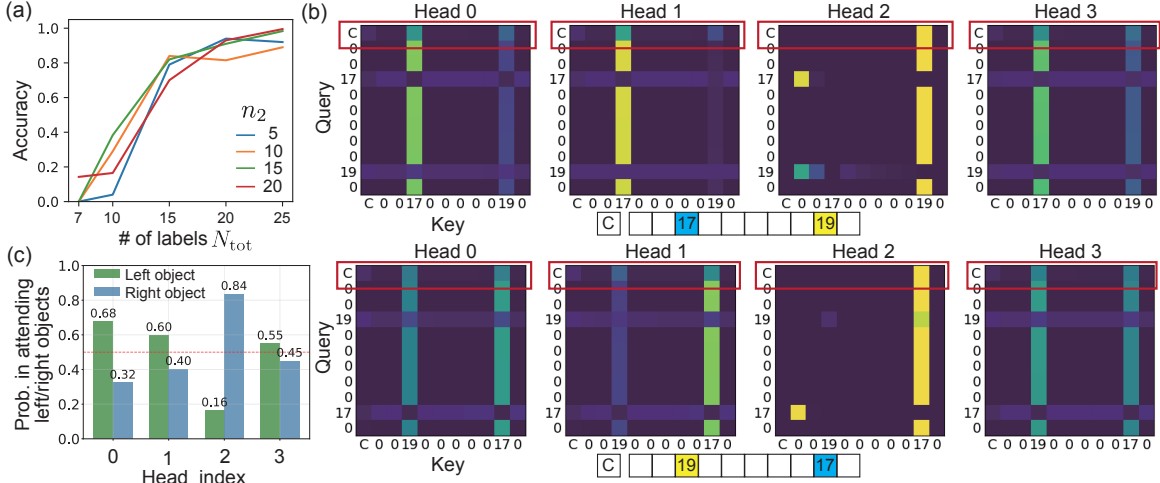

*Figure 4.* Analysis of reduced 1-layer model with $M_B = M_{\mathrm{rep}} = 1, M_h = 4$. The other parameters are the same as those used in Fig. 3. (a) Accuracy for unseen-pair generalization. (b) Examples of the attention pattern of the vision encoder. "C" in the 1D image corresponds to the class token. The red rectangles highlight the class token row in each attention map. The model trained with $N_{\mathrm{tot}} = 20, N_{\mathrm{pair}} = 15, n_2 = 10$ are used for the visualization. (c) Probability that each head attends to the left or right object, computed from the class token's attention weights. The red dashed line indicates random guess 0.5. This analysis is performed on images containing label pairs not seen during training (*i.e.* pairs formed from labels 16–20).

## 5. Coupling of Token and Positional Embeddings Induces Attention Gradients

We analyze how the vision encoder's attention becomes left–right asymmetric by expanding the pre-softmax attention logits into interpretable components. We decompose the pre-softmax attention logits $QK^T$ into weight and bias terms for each head:

$$QK^T = XW_{QK}X^T + XW_Q^T B_K + B_Q^T W_K X^T$$
$$+ B_Q^T B_K, \text{ where } W_{QK} = W_Q^T W_K. \quad (1)$$

With this weight-bias decomposition, we find that the term $XW_{QK}X^T$ dominates the contribution to the logits (See App. E), so we analyze it further. Writing $X$ as the sum of token embeddings $E$ and positional embeddings $P$, we expand $XW_{QK}X^T$ into four components:

$$XW_{QK}X^T = EW_{QK}E^T + EW_{QK}P^T + PW_{QK}E^T$$
$$+ PW_{QK}P^T. \quad (2)$$

Figure 5(a) shows the positional-token embedding decomposition for head 2 on the 1D image from Fig. 4(b) (App. F for the remaining heads). Interestingly, the cross term $EW_{QK}P^T$ that involves positional embeddings exhibits a clear horizontal gradient in the attention logits. This gradient creates a systematic rightward bias, assigning greater attention to the object on the right. In models that fail to generalize, this gradient is absent (App. G), implicating positional-embedding–driven asymmetry as the mechanism that enables generalization. As shown in Fig. 5(b), the position-independent (label-specific) contribution $EW_{QK}E^T$ yields

a larger contribution for label 19 than for label 17. Nevertheless, the right-biased cross term $EW_{QK}P^T$ causes the right-hand object to receive more attention even when the two objects' positions are swapped.

To quantify this effect, we compute, for each two-object image with an unseen label pair, the difference in CLS→object logits between the right and left objects from the two components: $\Delta_{\mathrm{label}}(\Delta_{\mathrm{p.e.}})$ is the difference of $EW_{QK}E^T(EW_{QK}P^T)$ at the object positions (Fig. 5(c)). We classify a head as relational when $|\Delta_{\mathrm{label}}| < |\Delta_{\mathrm{p.e.}}|$ (with the sign of $\Delta_{\mathrm{p.e.}}$ matching the geometric direction); otherwise it is label-specific. In Fig. 5(d), we show the probability that each head attends to the two objects in a relational or label-specific manner. These probabilities are quantitatively consistent with Fig. 4(c): the difference in the probability of attending to the left versus right object is approximately equal to the probability that each head attends to the two objects in a relational manner. This indicates that the left–right attention bias arises primarily from the monotonic gradient in the EP term.

We then ablate this effect by zeroing the positional-embedding contributions to the attention during inference and re-evaluating. In addition to terms from $XW_{QK}X^T$, we also ablate the BP term $B_Q^T W_K P^T$, which arises from $B_Q^T W_K X^T$ in Eq. 1, as it can induce position-specific attention (see App. E). As shown in Fig. 5(e), ablating the EP term most strongly suppresses left–right discrimination accuracy, indicating that the horizontal gradient induced by this term is essential for left–right discrimination. The drop in accuracy to near 0.5 indicates that the ablated model can

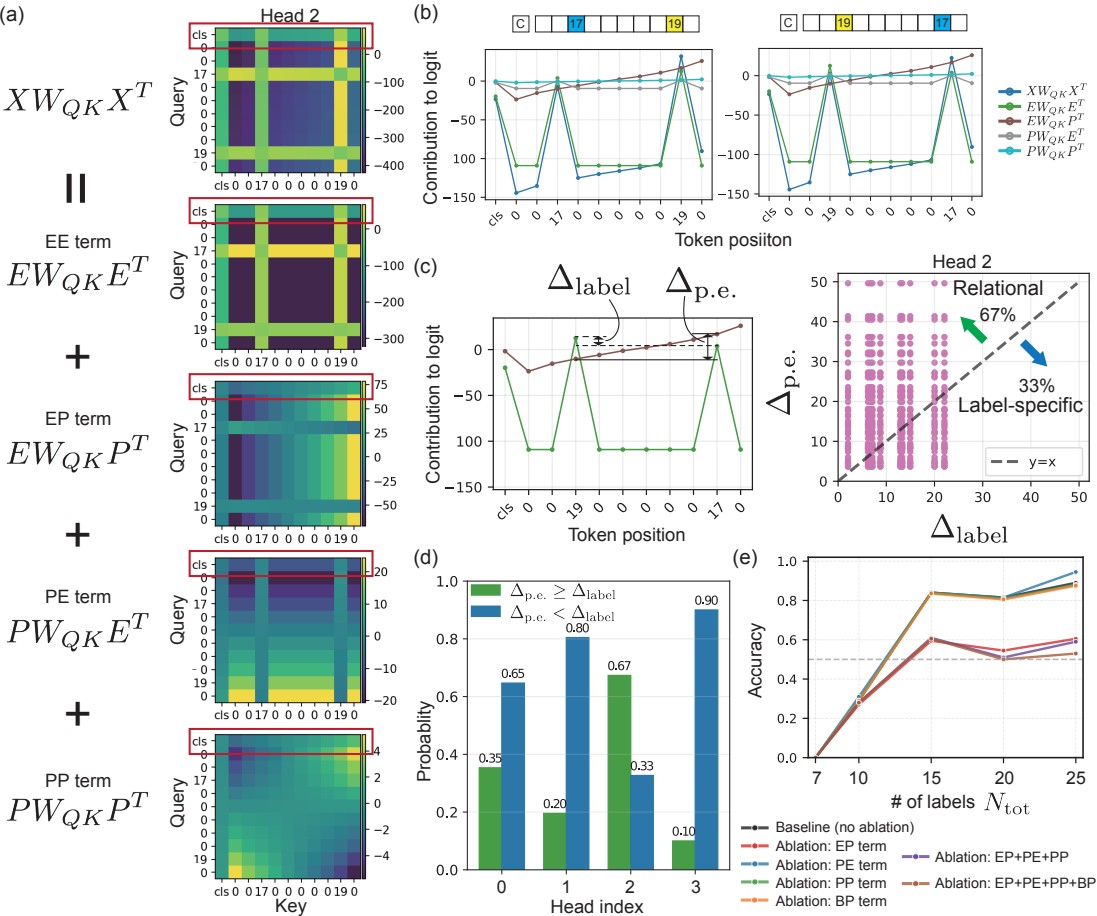

*Figure 5.* Attention gradient emerging in the attention pattern. (a) Contribution of decomposed terms to logit of the attention is shown for head 2 as a representative example. The red rectangles highlight the class token row in each attention map. The model for Fig. 4(b,c) is used for this analysis (a-d). (b) Spatial profiles of the total and each component of the class token row for two images with swapped object positions; colors denote the total and the four decomposed terms. (c) Definition of $\Delta_{\text{p.e.}}$ and $\Delta_{\text{label}}$ is schematically shown. $\Delta_{\text{p.e.}}$ is plotted against $\Delta_{\text{label}}$ for head 2. Each scatter point represents a test 1D image. (d) The probability that each head attends to the two objects based on their spatial relationship ($|\Delta_{\text{p.e.}}| < |\Delta_{\text{label}}|$) versus in a label-specific manner ($|\Delta_{\text{p.e.}}| < |\Delta_{\text{label}}|$). (e) Effect of ablating positional-embedding–derived logit components. Accuracy for unseen object pairs generalization is shown for different ablation conditions. Baseline is the model from Fig. 4(a) ($n_2 = 10$). At inference, we zero specific pre-softmax attention logit terms for all four heads: EP, PP, PE terms and the BP term $B_Q^T W_K P^T$.

recognize which objects appear together, but fails to encode their spatial relationship (App. H). In the App. I, we also ablate the position-dependent contribution from the VP term $PW_V^T$ to the value vector in the Transformer and find that the accuracy decreases to 0.5, indicating that the combination of attention and value likely enhances generalization.

Together, these results demonstrate that, in our toy setting, left–right discrimination in the vision encoder is predominantly mediated by an attention gradient arising from positional embeddings. We complement our empirical findings with a theoretical analysis (App. J) that explains why position-dependent attention asymmetry is necessary for generalization to unseen object pairs. Extending this analysis to Rotary Position Embedding (RoPE (Su et al., 2024)), widely used in modern language models such as Llama

3 (Grattafiori et al., 2024), Qwen3 (Yang et al., 2025), and Gemma 3 (Team et al., 2025), and also in vision transformer (Heo et al., 2024), we identify alternative mechanisms by which RoPE can induce the required positional bias (App. J).

We further verify that this mechanism is not specific to the 1D setting by extending our experiments to 2D images. We place two objects in $4 \times 4$ images, assign relational texts based on their horizontal positions, and flatten the image into a sequence of 16 tokens. The model generalizes to unseen object pairs (e.g., 98% unseen-pair generalization accuracy with $N_{\text{tot}} = 20, n_2 = 10$), and the attention decomposition reveals the same horizontal gradient from positional embeddings as in the 1D case (see App. K for details).

# 6. Alignment between Vision and Text Encoders

## 6.1. Attention Pattern in Text Encoder

Figure 6(a) visualizes attention patterns in the text encoder for the inputs "17 is on the left of 19" and "19 is on the left of 17." The text encoder is trained jointly with the vision encoder analyzed in Fig. 4(b) and shares the same 1-layer, 4-head architecture. For instance, head 3 exhibits a pronounced leftward bias in EOT→word-token attention: the subject entity (the first-mentioned object) receives the largest weight, independent of whether its label is 17 or 19.

We quantify these patterns by averaging the attention maps over unseen label pairs and reporting the mean and standard deviation (Figs. 6(b,c)). Head 3 is the most strongly left-biased, whereas heads 0, 1, and 2 show rightward bias.

Mirroring the vision encoder, the text encoder thus exhibits left–right symmetry breaking: directional structure in attention allows the model to identify which object is first in the sentence. Here we focus on a 1-layer text Transformer, where only EOT→word attention contributes to the final EOT representation; in deeper models, the causal mask naturally induces inherent left–right asymmetry at each layer, and we expect similar directional effects is enhanced.

## 6.2. Alignment between Image and Text Representation Space

We examine the relationship between the image and text representation spaces. One might expect that, in a CLIP-trained model, token embeddings for the same label align across the image and text spaces. However, raw cosine similarity reveals little direct alignment (Fig. 7(a)). We hypothesize that the two spaces are related up to a rotation. Fitting a rotation matrix to align a subset (label: $1 - 15$) of token embeddings associated with vision and text encoders and evaluating on a validation set (label: $16 - 20$), we observe markedly stronger alignment (Fig. 7(b)). This suggests that token representations in the text and image encoders align modulo a rotational degree of freedom—plausibly implemented by progressive rotations across Transformer layers—culminating in image–text alignment at the model output.

# 7. Generalization in the Presence of Both Left and Right Textual Representations

So far, we have considered only the *left* textual representation. Here, we investigate whether introducing a complementary *right* textual representation still enables a CLIP-style Transformer to learn object positional relations. To evaluate this, given an image, we check whether the top-2 most similar texts correspond to the two equivalent descrip-

tions of the correct spatial relationship—e.g., "X is on the left of Y" (XLY) and "Y is on the right of X" (YRX).

We train both the vision and text encoders as 1-layer, 4-head models. In this setting, the model does not generalize: at evaluation, the top-2 image–text similarities include sentences containing the correct object pair (XLY, XRY, YLX, or YRX), but the two semantically equivalent descriptions of the correct spatial relationship (e.g., both XLY and YRX) do not both appear among them (App. L). This empirical observation suggests that single-layer text encoder architectures have insufficient capacity for learning relational structure. To increase representational capacity, we add a second layer to the text encoder while keeping the vision encoder at 1 layer, and retrain. As in the "left-only" setting, increasing the variation in training labels leads to systematic generalization (Fig. 8(a,b)).

As shown in Fig. 8(c,d), analysis of the vision encoder's attention reveals the emergence of left- and right-biased heads, similar to those observed in the "left-only" setting (Sec. 4, 5). Ablation experiments confirm that the EP term remains the dominant causal driver of spatial discrimination in this setting, consistent with the left-only case (see App. M). This suggests that the model encodes relative positions in a manner analogous to the "left-only" case. When both left and right textual representations coexist, how the text encoder discriminates among XLY, XRY, YLX, and YRX remains unclear due to the complexity in interpreting 2-layer Transformer model. Nevertheless, we observe attention gradients in the vision encoder that break left–right symmetry, enabling the model to capture spatial relations.

# 8. Related Work

## 8.1. CLIP-Like VLMs and Autoregressive VLMs

CLIP casts image–text learning as large-scale contrastive alignment with dual encoders, enabling strong zero-shot transfer and retrieval (Radford et al., 2021). Follow-ups establish reproducible scaling laws and emphasize data/compute quality, while also noting compositional and spatial limits (Cherti et al., 2023; Yuksekgonul et al., 2023). CLIP-like variants broaden objectives and architectures: BLIP adds image–text matching and captioning with boot-strapped supervision (Li et al., 2022); ALIGN scales contrastive pretraining (Jia et al., 2021); ALBEF fuses modalities (Li et al., 2021); SLIP adds self-supervised vision loss (Mu et al., 2021); FILIP targets token-level alignment (Yao et al., 2022); and CoCa unifies contrastive and captioning losses (Yu et al., 2022). Autoregressive VLMs pair visual encoders with LLMs for open-ended generation, often using CLIP-based vision encoders: SimVLM adopts a prefix-LM objective (Wang et al., 2022), and LLaVA aligns vision-language models via multimodal instruction

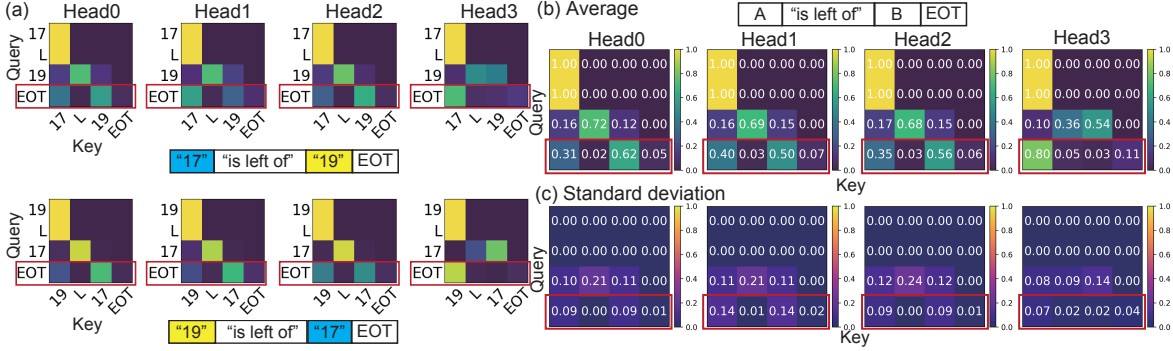

Figure 6. Attention patterns of encoder. (a) Representative examples with object 17 and 19 with swapped spatial relations are shown. (b-c) The average and the standard deviation of the attention map averaged over the text samples are shown. The text encoder is trained jointly with the vision encoder analyzed in Fig. 4(b) ($N_{\text{tot}} = 20, n_2 = 10$). The red rectangles highlight EOT's row in each attention map.

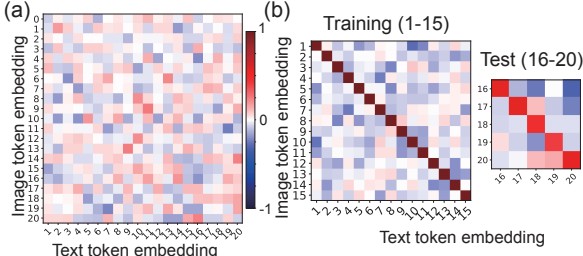

Figure 7. Aligning image and text token embeddings via learned rotation matrices. (a) Cosine similarity between image and text token embeddings for each label prior to optimization. (b) A rotation matrix is optimized to align text token embeddings with their corresponding image token embeddings for labels 1–15 (training). The learned rotation matrix is then evaluated on labels 16–20 (test).

tuning (Liu et al., 2023; 2024).

## 8.2. Compositional Understanding in VLMs

VLMs excel at category recognition but remain brittle on spatial reasoning and compositional generalization, with performance dropping when language priors and dataset shortcuts are controlled (Kamath et al., 2023; Tong et al., 2024; Yuksekgonul et al., 2023). A growing body of work has developed benchmarks to evaluate these failures, including spatial relations, across diverse settings (Parcalabescu et al., 2022; Zhao et al., 2022; Hsieh et al., 2023; Kamath et al., 2023; Rajabi & Kosecka, 2024). A central challenge is attribute/role binding: models often misattach properties or relations to the wrong object in multi-entity scenes (Yuksekgonul et al., 2023; Campbell et al., 2024; Lewis et al., 2024; Newman et al., 2024). Negation further exposes shallow matching, yielding high false positives without targeted counterfactuals (Alhamoud et al., 2025; Singh et al., 2025). While these works provide valuable characterizations of where VLMs succeed and fail, they focus primarily on *evaluating* spatial and compositional capabilities rather than explaining the internal mechanisms that give rise to them.

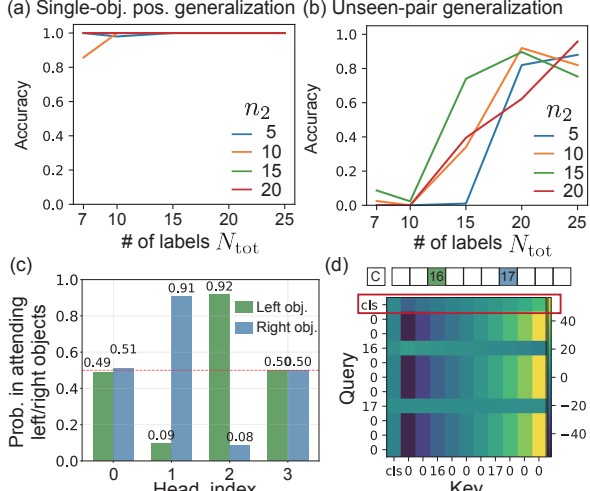

Figure 8. Generalization in the setup with left and right textual representation. (a,b) Accuracy is shown for single-object positional and unseen-pair generalization. The model parameters are $M_B^{\text{vis}} = 1, M_{\text{rep}}^{\text{vis}} = 1, M_h^{\text{vis}} = 4, M_B^{\text{txt}} = 2, M_{\text{rep}}^{\text{vis}} = 1, M_h^{\text{txt}} = 4, d_{\text{head}} = 32, d_{\text{model}} = 128$. (c) Probability that each head attends to the left or right object, computed from the class token's attention weights. The red dashed line indicates random guess 0.5. (d) Contribution from the EP term to logit of the attention of head1 is shown for a 1D image with label 16 and 17. The analyzed model in (c,d) is with the parameters $N_{\text{tot}} = 20, n_2 = 10$. The red rectangle highlights the class token row in each attention map.

We note that attribute binding for single objects (e.g., associating "yellow" with "circle") and relational understanding between multiple objects (e.g., "A is left of B") are likely distinct problems requiring different mechanisms: the former involves associating properties with an individual object, while the latter requires comparing positions across objects. To our knowledge, none of the prior works identify a specific mechanistic pathway responsible for spatial relation learning in VLMs, nor show that ablating it directly degrades performance. We contribute a controlled setting that offers a minimal mechanistic account of one such relational

ability: left–right discrimination, complementing existing evaluative studies with mechanistic understanding that has been largely unexplored.

## 9. Discussion & Conclusion

We construct a paired dataset of 1D images with one or two objects and textual descriptions of their spatial relations, and show that CLIP-style training enables a model to acquire an understanding of left–right relations and to generalize to unseen object pairs. By analyzing attention in models that generalize, we find that the interaction between positional and token embeddings induces a horizontal gradient in the attention map, breaking left–right symmetry. This emergent asymmetry provides a plausible mechanism for relational discrimination in our setting. While recent work has shown fundamental geometric limitations of CLIP's latent space for spatial reasoning (Kang et al., 2025), the mechanisms by which CLIP-style models can nonetheless learn to discriminate spatial relations remain unclear. We address this gap by identifying the critical role of attention gradients from positional embeddings.

Our finding that label diversity drives generalization connects to recent work on data diversity in compositional generalization (Uselis et al., 2025). While both works observe that diversity is key, the types of generalization may involve different mechanisms: in Uselis et al. (2025), diversity forces an additive factorization of context-free attributes (e.g., color and shape), whereas in our setting, diversity appears to force a position-dependent attention mechanism to generalize across object identities. This may reflect a difference between attributive and relational understanding: spatial relations like "left of" are not properties of individual objects but comparisons between positions, which may require mechanisms beyond additive decomposition. Our ablation experiments (Fig. 5(e)) are consistent with this view, showing that removing the positional embedding contributions eliminates spatial discrimination while preserving object recognition. Furthermore, we extend our experiments to 3-object images, where a simple additive role assignment (e.g., "left object" and "right object") is no longer sufficient. The same attention gradient mechanism generalizes to this setting (see App. N), providing further evidence that the model relies on positional comparison rather than role-conditioned codes.

Our deliberately simplified setup leaves open whether similar attention gradient mechanisms emerge in large-scale VLMs trained on natural 2D images, where richer spatial structure and compositionality arise. While we observe the same mechanism in a 2D toy setting (see App. K), bridging to natural images remains an important goal. While we focus here on CLIP-style contrastive training, autoregressive VLMs are increasingly prevalent. Although we interestingly

observe a similar attention gradient mechanism in an autoregressive VLM setting (see App. O), a more thorough comparison of spatial relation competence across training paradigms remains a natural next step. Our analysis also leaves open how multiple heads interact (App. P) and how relational signals propagate across layers (App. Q). As a preliminary step toward analyzing non-linear models, we propose an approximate decomposition of LayerNorm that separates positional and token embedding contributions. We find that the attention gradient persists in the positional-embedding-related terms of a multi-layer non-linear model (see App. R). Extending this analysis to fully characterize the role of non-linear components remains a significant challenge.

Nevertheless, our results suggest a clear link between positional attention gradients and relational generalization, providing what may be viewed as a first-order mechanistic understanding of spatial discrimination in VLMs. This follows a well-established approach in mechanistic interpretability, where controlled minimal settings have yielded insights that later generalized to larger models (Elhage et al., 2021; Olsson et al., 2022). We hope that this baseline understanding can serve as a foundation for progressively incorporating the effects of non-linear components, deeper layers, and richer spatial structure. We also hope that this work may inform the design of models that reliably encode spatial relations, for instance, by highlighting the role of positional embeddings and the minimum architectural depth required for symmetry breaking.

## Impact Statement

This paper presents mechanistic research on how CLIP-style vision–language models acquire relational understanding, using a minimal synthetic testbed. As foundational work focused on interpretability and understanding, there are no specific ethical concerns or societal consequences that we feel must be highlighted here.

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

## A. Overview of Dataset Splits and Generalization Types

In Fig. 9, we illustrate the training and test datasets used to evaluate the three types of generalization defined in the main text. For clarity, we depict a simplified setting with a single image per label (single-object) and per ordered label pair (two-object): $D_{\text{image}} = 10$, $n_1 = n_2 = 1$, $N_{\text{tot}} = 5$, $N_{\text{pair}} = 3$, and $N_{\text{val}} = N_{\text{tot}} - N_{\text{pair}} = 2$.

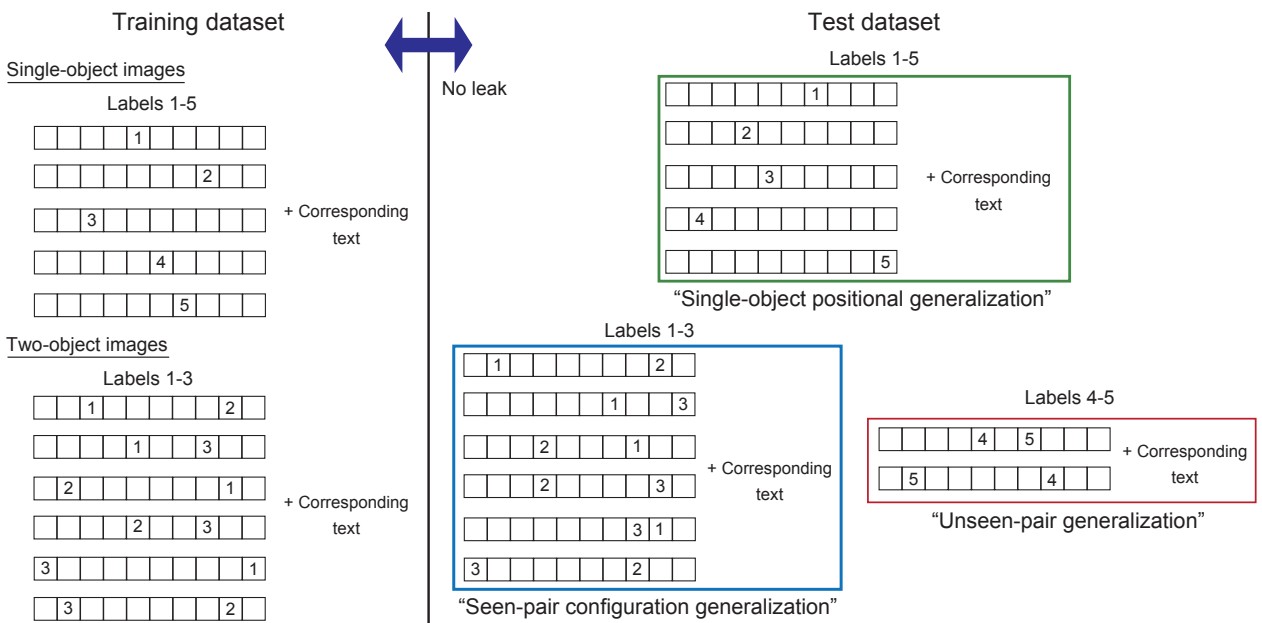

*Figure 9.* Schematic of training and test datasets for evaluating three types of generalization. Single-object images are used to assess single-object positional generalization, while two-object images are used to assess seen-pair configuration generalization and unseen-pair generalization. For clarity, we depict a simplified setting with one image per label or ordered label pair ($n_1 = n_2 = 1$), $N_{\text{tot}} = 5$ object categories, $N_{\text{pair}} = 3$ categories used for two-object training, and $N_{\text{val}} = 2$ held-out categories for unseen-pair evaluation.

## B. Details of Model Training

### B.1. Compute Resources

All experiments were conducted on NVIDIA A100, H100, and H200 GPUs on internal servers. Training a single model for 10,000 epochs requires at most approximately 12 hours of wall-clock time and 4GB of GPU memory for the largest experiments in the main text (Sec. 3, 7). We estimate the total compute for the main text experiments to be approximately 720 GPU hours.

### B.2. Public Codes

Our implementation builds upon the following publicly available codebases, which we modified for our experimental setting:

- **CLIP**: https://github.com/openai/CLIP

- **Transformer**: https://github.com/Sea-Snell/grokking

## C. Effect of Weight Decay on the Generalization

In Fig. 10, we examine how the weight decay regularization affects the unseen-pair generalization. The model is the same architecture with those studied in Sec. 2.3.

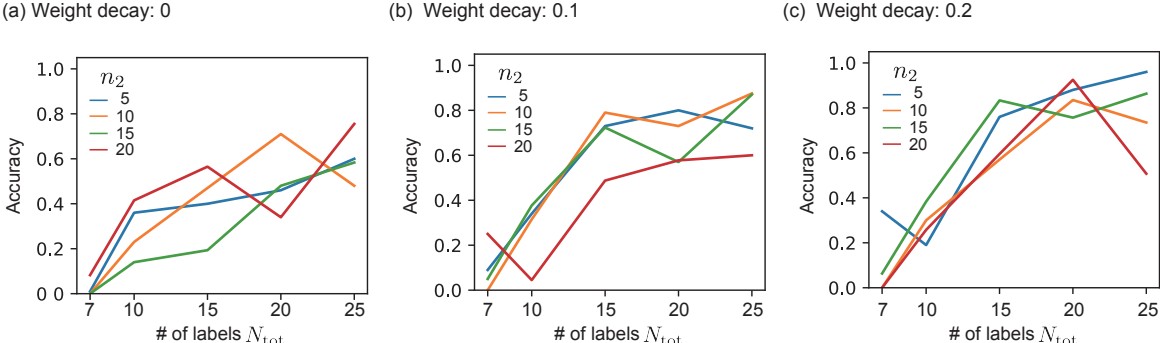

*Figure 10.* Unseen-pair generalization is analyzed for the models trained with different weight decay $w$. $M_B = 2, M_{\text{rep}} = 2, M_h = 4, d_{\text{head}} = 32, d_{\text{MLP}} = 512, d_{\text{model}} = 128$.

# D. Generalization Dynamics

Figure 11 shows the dynamics of generalization accuracy and training loss during training. We find that single-object positional, seen-pair configuration, and unseen-pair generalization are achieved consecutively. Based on the completion of single-object positional and seen-pair configuration generalization, we define three phases of training. As shown in Fig. 11(a,c), in Phase 1, single-object positional generalization is achieved; in Phase 2, seen-pair configuration generalization is achieved; and in Phase 3, unseen-pair generalization is enhanced. We observe pronounced jumps in training loss near the phase-1/phase-2 boundary, whereas any analogous signature at the phase-2/phase-3 boundary is weak or absent. The underlying mechanism remains unclear at present.

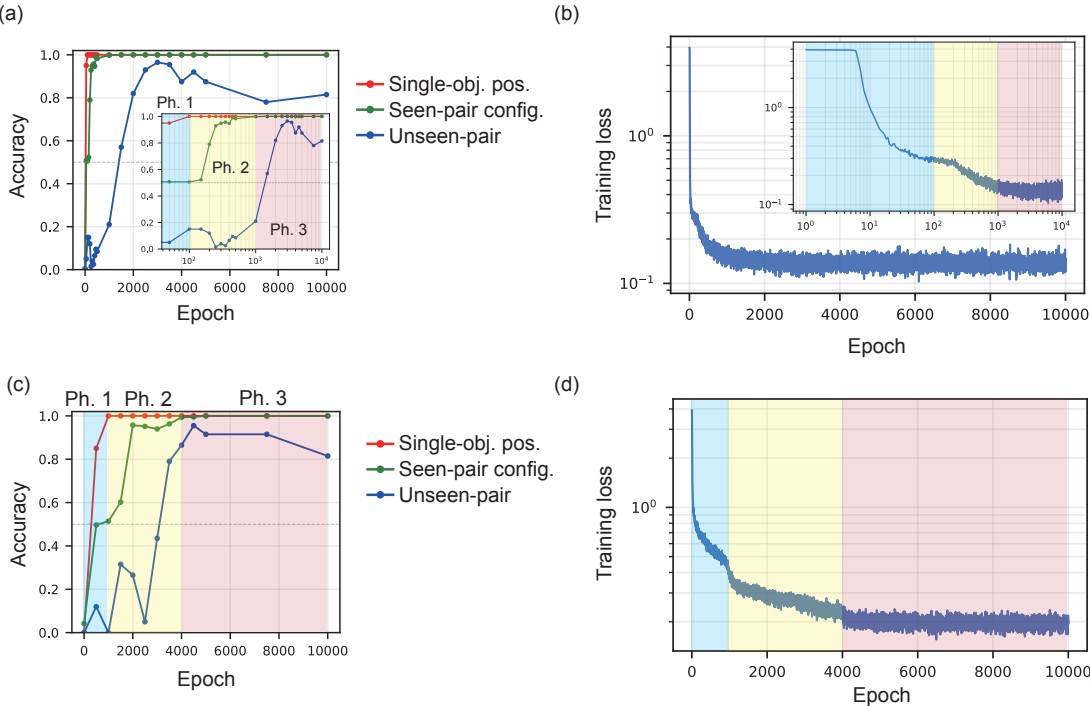

*Figure 11.* Generalization dynamics. (a,c) Dynamics of accuracy for three types of generalization: single-object positional, seen-pair configuration, unseen-pair generalization defined in the main text. By the completion of single-object positional and seen-pair configuration generalization, we define three phases in the training. In (b,d), we show the corresponding dynamics of the training loss. In (a,b), the dynamics is shown for the model analyzed in Fig. 3 ($M_B = M_{\text{rep}} = 2, M_h = 4, N_{\text{tot}} = 20, n_2 = 10$). In (c,d), dynamics is shown for the model analyzed in Fig. 4 ($M_B = M_{\text{rep}} = 1, M_h = 4, N_{\text{tot}} = 20, n_2 = 10$).

# E. Weight-Bias Decomposition of Pre-Softmax Logit for All the Heads of Generalizing Model

Figure 12 shows the weight-bias decomposition of pre-softmax logit $QK^T$ for all four heads for the model analyzed in Fig. 5.

For a 1-layer vision Transformer, the output representation of the class token is determined by its attention distribution over image pixels. Since softmax is shift-invariant, only terms that vary across key positions (columns) affect this attention. The terms $XW_Q^T B_K$ and $B_Q^T B_K$ are constant across columns for the class token row and thus do not contribute to attention discrimination. The remaining terms—$XW_{QK}X^T$ and $B_Q^T W_K X^T$—vary across columns and determine the attention pattern. To compare the relative strength of these discriminative terms, we compute the standard deviation across the class token row of each term's contribution to the logit matrix. The result is shown in Fig. 13. The average standard deviation of the $XW_{QK}X^T$ term accounts for 76%, 77%, 91%, and 76% of the total standard deviation of $QK^T$ for the four heads, respectively.

In Fig. 12, we observe a horizontal gradient in $B_Q^T W_K X^T$ similar to that in $EW_{QK}P^T$ (Sec. 5). This term can be decomposed as $B_Q^T W_K X^T = B_Q^T W_K E^T + B_Q^T W_K P^T$. The $B_Q^T W_K P^T$ component can therefore induce position-dependent attention, analogous to $EW_{QK}P^T$.

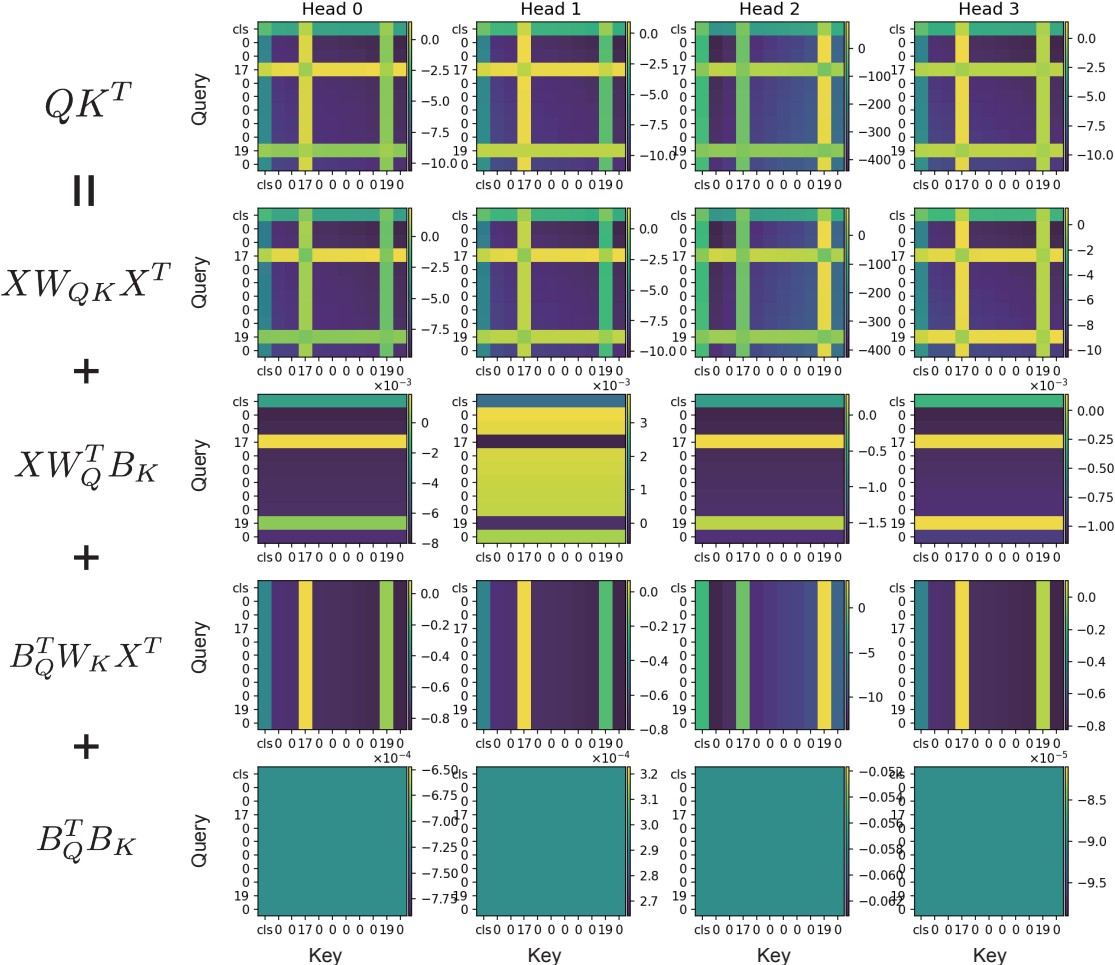

*Figure 12.* Weight-bias decomposition of pre-softmax logit for all four heads for the model analyzed in Fig. 5.

*Figure 13.* The standard deviation across the class token row of each term's contribution to the logit matrix. Each count corresponds to a single 1D image.

## F. Positional-Token Embedding Decomposition of Pre-Softmax Logit for All the Heads of Generalizing Model

Figure 14 shows the per-head positional-token embedding decomposition of $XW_{QK}X^T$ term for all four heads on the 1D image from Fig. 4(b). Clear monotonic pattern along the horizontal is observed among all the four heads. The direction of the gradient is consistent with the bias in the attention to left and right objects investigated in Fig. 4(c).

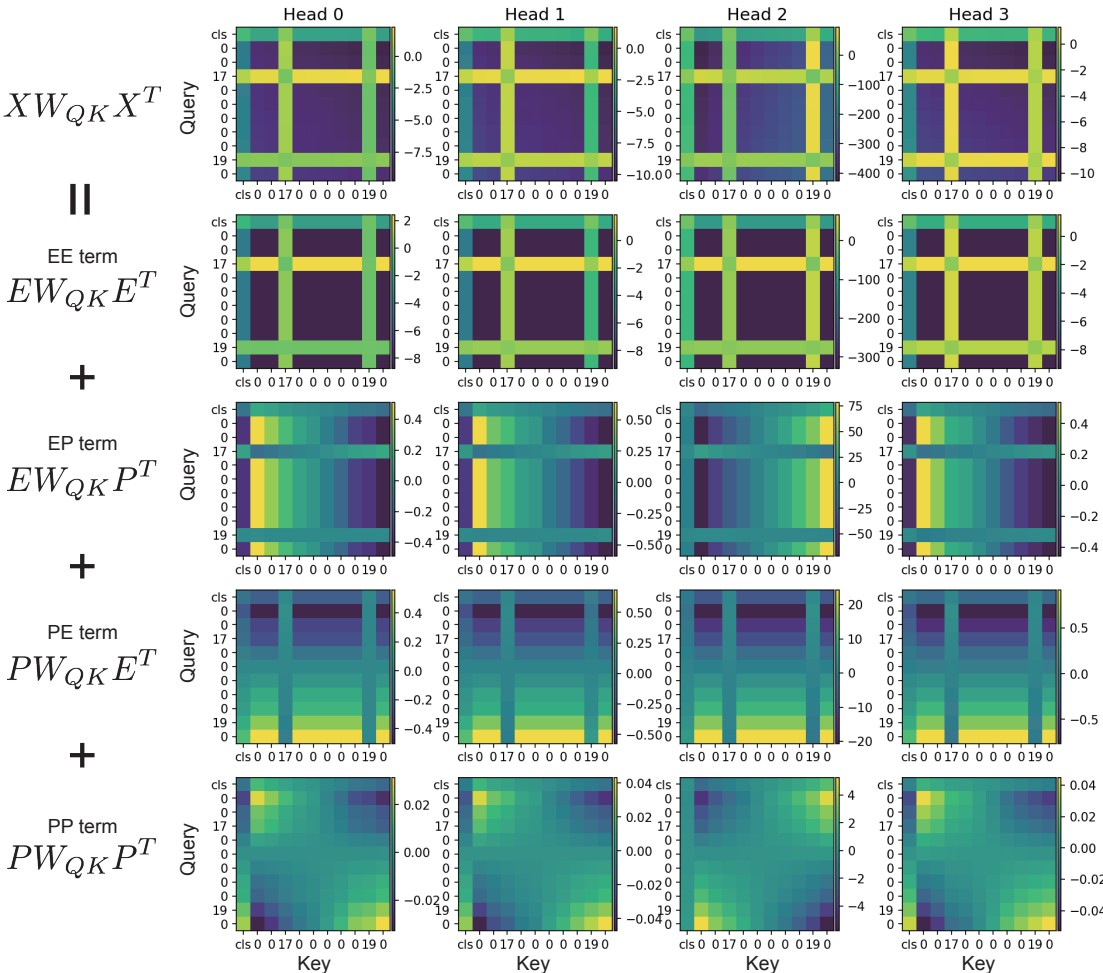

*Figure 14.* Positional-token embedding decomposition of pre-softmax logit for all the heads of generalizing model. The model is that analyzed in Fig. 5.

## G. Absence of Positional–Token Attention Gradients in a Non-Generalizing Model

Figure 15(a) reports left–right attention bias for each of the four heads in a model that shows little generalization ($N_{\text{tot}} = 10, n_2 = 5$ in Fig. 4(a)). For each test image, we identify which token (left object, right object) receives the maximal attention the from CLS token for a given head, and compute the proportion over the test set as performed in Sec. 4. No head exhibits a consistent left or right bias: proportions are near chance and vary across labels, indicating label-specific, non-relational attention. Notably, for head 3, the probabilities for both left and right are low, implying that the CLS token frequently attends most to background tokens (label 0) rather than to either object.

Figure 15(b) plots the EP cross term contribution to logits for all heads. The profiles at the CLS row show no clear monotonic pattern along the horizontal axis (no left-to-right or right-to-left gradient). This absence of a positional–token–induced attention gradient coincides with the lack of generalization, consistent with our finding that such gradients are necessary for robust left–right discrimination.

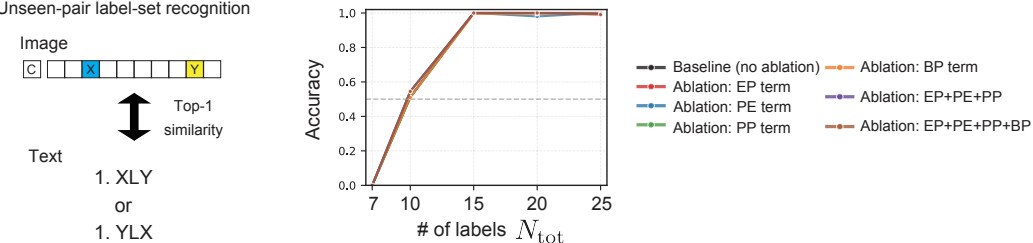

*Figure 15.* Absence of positional–token attention gradients in a non-generalizing model. (a) Probability that each head attends to the left or right object the most, computed from the class token's attention weights. (b) Spatial profiles of the EP term defined in the main text for all heads.

## H. Unseen-Pair Label-Set Recognition in Two-Object Images

In Fig. 5(e), we show that the accuracy for unseen-pair generalization drops to near 0.5. Here, we further investigate the models to understand what they are capable of in this regime. Figure 16 shows the accuracy with which the model identifies the two objects present in two-object images under different ablation conditions. We consider the model's prediction correct if the caption for a two-object image containing objects X and Y is predicted as either XLY or YLX. This new metric confirms that even when the accuracy for unseen-pair generalization drops to near 0.5 in Fig. 5(e), the ablated model can still recognize which objects appear together in two-object images.

*Figure 16.* Accuracy of unseen-pair label-set recognition in two-object images under different ablation conditions. The label pairs not seen during training is used in this test. Baseline is the model from Fig. 4(a) ($n_2 = 10$). At inference, we zero specific pre-softmax attention logit terms for all four heads: EP, EP, PP terms and the BP term $B_Q^T W_K P^T$.

## I. Effect of Position-Dependent Contribution of Value Vector in Transformer on Generalization

The value $V = XW_V^T + B_V$ can be decomposed into $V = EW_V^T + PW_V^T + B_V$. Among these terms, the VP term $PW_V^T$ is the position-dependent contribution to the value, which may affect generalization in recognizing the relative positions of two objects in two-object images. We therefore investigate the effect of this term through ablation, as shown in Fig. 17.

In Fig. 17(a), we find that the accuracy for unseen-pair generalization decreases to 0.5 when we ablate all position-dependent terms from both attention and value. In contrast, as shown in Fig. 17(b), the accuracy of unseen-pair label-set recognition defined in Sec. H is still kept under the ablation. This indicates that, although the EP term is dominant for left–right discrimination in the vision encoder, the combination of attention and value enhances unseen-pair generalization.

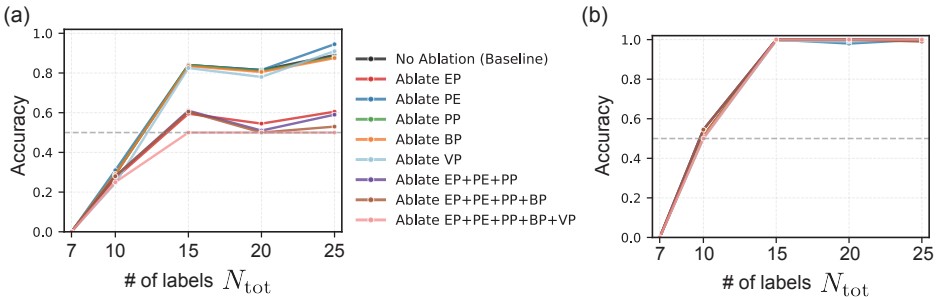

*Figure 17.* Effect of ablating positional-embedding–derived logit components. (a) Accuracy for unseen-pair generalization is shown for different ablation conditions. Baseline is the model from Fig. 4(a) ($n_2 = 10$). At inference, we zero specific pre-softmax attention logit terms for all four heads: EP, PE, PP, BP and VP terms. (b) Accuracy of unseen-pair label-set recognition in two-object images under different ablation conditions. The label pairs not seen during training is used in this test.

# J. Positional Encoding and Relational Generalization

We theoretically analyze how different positional encoding schemes—learnable positional embeddings and Rotary Position Embedding (RoPE (Su et al., 2024))—enable out-of-distribution generalization to unseen object pairs in 1-layer transformer models.

## J.1. Analysis with Learnable Positional Embeddings Only

We first analyze the case without RoPE, using only learnable positional embeddings, which corresponds to the experiments in the main text. Consider a 1-layer vision transformer with a CLS token. For an image with object $X$ at position $a$ and object $Y$ at position $b$ (where $a < b$), the attention mechanism computes queries and keys via learned projections $W_Q, W_K \in \mathbb{R}^{d_{\text{head}} \times d_{\text{model}}}$ with bias vectors $b_Q, b_K \in \mathbb{R}^{d_{\text{head}}}$.

Following the convention $Q = XW_Q^T + b_Q$, for single tokens as row vectors, the query from the CLS token $C \in \mathbb{R}^{d_{\text{model}}}$ (which includes its positional embedding) and keys for each object are:

$$q = CW_Q^T + b_Q \in \mathbb{R}^{d_{\text{head}}} \tag{3}$$

$$k_x = (E_x + P_a)W_K^T + b_K \in \mathbb{R}^{d_{\text{head}}} \tag{4}$$

$$k_y = (E_y + P_b)W_K^T + b_K \in \mathbb{R}^{d_{\text{head}}} \tag{5}$$

where $E_x, E_y \in \mathbb{R}^{d_{\text{model}}}$ are token embeddings, and $P_a, P_b \in \mathbb{R}^{d_{\text{model}}}$ are learnable positional embeddings.

The attention logits from CLS to each object are computed as the inner product:

$$S_x = \langle q, k_x \rangle = (CW_Q^T + b_Q)(W_K(E_x + P_a)^T + b_K^T) \tag{6}$$

$$S_y = \langle q, k_y \rangle = (CW_Q^T + b_Q)(W_K(E_y + P_b)^T + b_K^T) \tag{7}$$

**Condition for Consistent Directional Preference.**  In the main text, we show that the attention bias in the left and right objects is essential for the recognition of relationship between unseen pair of objects in Figs. 4 and 5. For the model to consistently attend more to the left (or right) object regardless of object identity, we require that swapping object positions preserves the relative attention preference. Let $S'_x$ and $S'_y$ denote the attention logits when $X$ and $Y$ are swapped (i.e., $Y$ at position $a$, $X$ at position $b$). The condition for consistent directional preference is:

$$(S_x - S_y)(S'_y - S'_x) > 0 \tag{8}$$

If the model always prefers the left object:

- Original configuration: $X$ is left $\Rightarrow S_x > S_y \Rightarrow (S_x - S_y) > 0$

- Flipped configuration: $Y$ is left $\Rightarrow S'_y > S'_x \Rightarrow (S'_y - S'_x) > 0$

**Decomposition into Token and Position Contributions.**  We decompose the attention difference into token-dependent and position-dependent terms. Note that the bias terms $b_K$ cancel in the difference:

$$S_x - S_y = (CW_Q^T + b_Q)W_K(E_x + P_a - E_y - P_b)^T \tag{9}$$

Denoting $\tilde{C} = CW_Q^T + b_Q$ for convenience:

$$S_x - S_y = \underbrace{\tilde{C}W_K(E_x - E_y)^T}_{T \text{ (token-dependent)}} + \underbrace{\tilde{C}W_K(P_a - P_b)^T}_{\Pi \text{ (position-dependent)}} \tag{10}$$

Similarly, for the flipped configuration:

$$S'_y - S'_x = \underbrace{\tilde{C}W_K(E_y - E_x)^T}_{-T \text{ (token-dependent)}} + \underbrace{\tilde{C}W_K(P_a - P_b)^T}_{\Pi \text{ (position-dependent)}} \tag{11}$$

Note that the token-dependent term flips sign $(-T)$ while the position-dependent term $\Pi$ remains the same. The consistency condition becomes:

$$(T + \Pi)(-T + \Pi) = \Pi^2 - T^2 > 0 \tag{12}$$

This simplifies to:

$$|\Pi| > |T| \tag{13}$$

**Requirement for OOD Generalization.** For unseen object pairs, the token-dependent term $T = \tilde{C}W_K(E_x - E_y)^T$ is unpredictable since the model has not learned specific interactions between these token embeddings and the $W_K$ projection. For the condition to hold robustly across all unseen pairs, we require:

$$|\Pi| \gg |T| \quad \text{for all unseen pairs} \tag{14}$$

That is, the position-dependent gradient $\Pi = \tilde{C}W_K(P_a - P_b)^T$ must dominate the token-dependent variation. This provides a theoretical justification for our empirical finding in the main text: the horizontal attention gradient induced by positional embeddings is essential for generalization to unseen object pairs.

### J.2. Experiments of 1-Layer Models with RoPE (No Learnable Positional Embeddings)

We examine whether RoPE alone, without learnable positional embeddings, can achieve relational generalization. Figure 18(a) shows results for a 1-layer model with RoPE ($M_B = M_{\text{rep}} = 1, M_h = 4$). We keep using the text encoder with learnable positional embeddings (no RoPE). Surprisingly, the model successfully generalizes to unseen object pairs, and exhibits a consistent left-right attention bias (Fig. 18(b,c)). This is notable because, without learnable positional embeddings, the attention gradient mechanism originating from the positional embedding propsoed in the main text cannot operate. We now analyze an alternative mechanism by which RoPE enables this generalization.

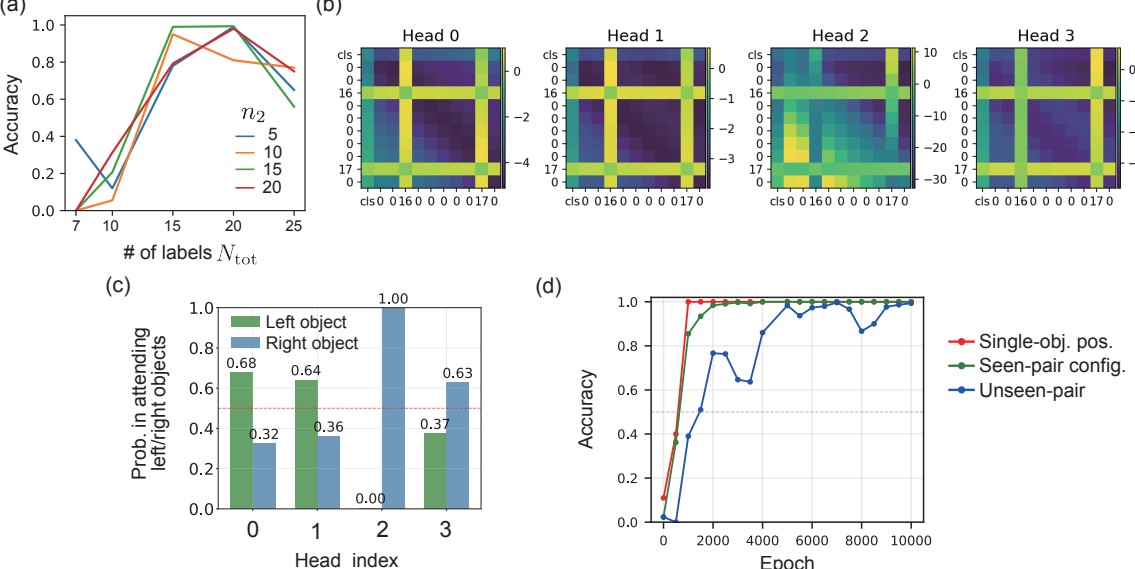

*Figure 18.* Analysis of reduced 1-layer model with RoPE (no learnable positional embedding, $M_B = M_{\text{rep}} = 1, M_h = 4$). The other parameters are the same as those used in Fig. 3. (a) Accuracy for unseen-pair generalization. (b) Examples of the attention pattern of the vision encoder. "C" in the 1D image corresponds to the class token. The model trained with $N_{\text{tot}} = 20, n_2 = 15$ are used for the visualization. (c) Probability that each head attends to the left or right object, computed from the class token's attention weights. The red dashed line indicates random guess 0.5. This analysis is performed on images containing label pairs not seen during training (*i.e.* pairs formed from labels 16–20). (d) An example of the generalization dynamics is shown for the model analyzed in (b,c).

## J.3. Analysis of RoPE without Learnable Positional Embeddings

We now extend the analysis to include RoPE with bias terms. Following standard implementations, RoPE rotations are applied after the linear projection. The query from the CLS token and keys for each object are:

$$q = CW_Q^T + b_Q \in \mathbb{R}^{d_{\text{head}}} \tag{15}$$

$$k_x = (E_x W_K^T + b_K) R_a^T \in \mathbb{R}^{d_{\text{head}}} \tag{16}$$

$$k_y = (E_y W_K^T + b_K) R_b^T \in \mathbb{R}^{d_{\text{head}}} \tag{17}$$

where $R_a = R(a \cdot \Delta\theta)$ is the RoPE rotation matrix. Note that the bias $b_K$ is also rotated by the position-dependent rotation.

The attention logits from CLS to each object are:

$$S_x = \langle q, k_x \rangle = (CW_Q^T + b_Q) R_a (W_K E_x^T + b_K^T) \tag{18}$$

$$S_y = \langle q, k_y \rangle = (CW_Q^T + b_Q) R_b (W_K E_y^T + b_K^T) \tag{19}$$

**Decomposition into Token, Position, and Bias Contributions.**   Denoting $\tilde{C} = CW_Q^T + b_Q$ for convenience, the attention difference becomes:

$$S_x - S_y = \tilde{C} R_a W_K E_x^T - \tilde{C} R_b W_K E_y^T + \tilde{C} R_a b_K^T - \tilde{C} R_b b_K^T \tag{20}$$

$$= \underbrace{\tilde{C}(R_a W_K E_x^T - R_b W_K E_y^T)}_{T_1 \text{ (token-dependent)}} + \underbrace{\tilde{C}(R_a - R_b) b_K^T}_{\Pi_{\text{bias}} \text{ (bias-position-dependent)}} \tag{21}$$

Similarly, for the flipped configuration:

$$S_y' - S_x' = \underbrace{\tilde{C}(R_a W_K E_y^T - R_b W_K E_x^T)}_{T_2 \text{ (token-dependent)}} + \underbrace{\tilde{C}(R_a - R_b) b_K^T}_{\Pi_{\text{bias}} \text{ (bias-position-dependent)}} \tag{22}$$

The consistency condition becomes:

$$(T_1 + \Pi_{\text{bias}})(T_2 + \Pi_{\text{bias}}) > 0 \tag{23}$$

### J.3.1. TWO MECHANISMS FOR OOD GENERALIZATION.

**Mechanism 1: Bias-Induced Positional Gradient.**   When $b_K \neq 0$, the term $\Pi_{\text{bias}} = \tilde{C}(R_a - R_b) b_K^T$ provides a position-dependent contribution analogous to the learnable positional embedding gradient $\Pi$ in the non-RoPE case. If $|\Pi_{\text{bias}}| \gg |T_1|, |T_2|$, the consistency condition is satisfied regardless of token identity. This mechanism emerges from the interaction between the bias term and position-dependent RoPE rotations.

**Mechanism 2: Learned Subspace Alignment (When $b_K = 0$).**   Even without bias terms, generalization can occur if $W_K$ learns to project all token embeddings into a low-dimensional subspace aligned with the RoPE rotation. As the simplest scenario, we consider the case where $W_K$ projects all tokens into a one-dimensional subspace:

$$W_K E_i^T \approx \alpha_i \mathbf{v} \quad \text{for all tokens } i \tag{24}$$

where $\mathbf{v} \in \mathbb{R}^{d_{\text{head}}}$ is a common unit vector and $\alpha_i > 0$ are token-specific positive scalars. Then:

$$T_1 = \alpha_x(\tilde{C} R_a \mathbf{v}) - \alpha_y(\tilde{C} R_b \mathbf{v}) \tag{25}$$

$$T_2 = \alpha_y(\tilde{C} R_a \mathbf{v}) - \alpha_x(\tilde{C} R_b \mathbf{v}) \tag{26}$$

Define the scalars $\rho_a = \tilde{C} R_a \mathbf{v}$ and $\rho_b = \tilde{C} R_b \mathbf{v}$, which represent the attention contribution from positions $a$ and $b$ respectively. Then:

$$T_1 = \alpha_x \rho_a - \alpha_y \rho_b \tag{27}$$

$$T_2 = \alpha_y \rho_a - \alpha_x \rho_b \tag{28}$$

The product becomes:

$$T_1 T_2 = (\alpha_x \rho_a - \alpha_y \rho_b)(\alpha_y \rho_a - \alpha_x \rho_b) \tag{29}$$

$$= \alpha_x \alpha_y (\rho_a^2 + \rho_b^2) - (\alpha_x^2 + \alpha_y^2)\rho_a \rho_b \tag{30}$$

Therefore, the condition $T_1 T_2 > 0$ is equivalent to:

$$\frac{\alpha_x \alpha_y}{\alpha_x^2 + \alpha_y^2} > \frac{\rho_a \rho_b}{\rho_a^2 + \rho_b^2} \tag{31}$$

The left-hand side is minimized when $\alpha_x/\alpha_y \to 0$ or $\infty$, approaching 0, and maximized at $\alpha_x = \alpha_y$ with value $1/2$. The right-hand side is similarly bounded in $(0, 1/2]$. When $\rho_a \gg \rho_b$ (strong positional asymmetry), the right-hand side approaches 0, making the condition easily satisfied for any positive $\alpha_x, \alpha_y$.

We summarize the mechanisms in Table 1.

*Table 1.* Summary of mechanisms.

| | Mechanism 1 (Bias-induced gradient) | Mechanism 2 (Subspace alignment) |
|---|---|---|
| Requires $b_K \neq 0$ | Yes | No |
| Requires low-rank $W_K$ | No | Yes |
| Positional gradient source | $\Pi_{\text{bias}} = \tilde{C}(R_a - R_b)b_K^T$ | $\rho_a \neq \rho_b$ via RoPE |

### J.3.2. ANALYSIS OF THE GENERALIZING MODEL.

We now verify that the above theory is consistent with the generalizing model analyzed in Fig. 18. Figure 19(a) (orange bars) shows the proportion of unseen object pairs satisfying the consistency condition (Eq. 23), which is quantitatively consistent with the left-right attention bias observed in Fig. 18(c).

**Mechanism 1: Bias-Induced Positional Gradient.** To examine whether Mechanism 1 operates in the trained model, we plot the bias-induced gradient $\Pi_{\text{bias}}$ as a function of the positional distance $b - a$ in Fig. 19(b). Since the RoPE rotation matrix is block-diagonal with each $2 \times 2$ block having a different rotation frequency (rather than a standard single-axis rotation), we show the mean and standard deviation for each disntance. We observe a clear positional dependence, confirming the contribution from Mechanism 1.

**Mechanism 2: Learned Subspace Alignment.** To isolate the contribution from Mechanism 2, we set $\Pi_{\text{bias}} = 0$ in Eq. 23 and evaluate whether the consistency condition $T_1 T_2 > 0$ still holds. Figure 19(a) (blue bars) shows that, even without the bias contribution, the consistency condition is largely satisfied, indicating that Mechanism 2 also contributes to the left-right attention bias.

To test our hypothesis that $W_K$ projects token embeddings into a low-dimensional subspace (Eq. 24), we perform principal component analysis (PCA) on the key vectors $\{W_K E_i^T\}$. Figure 19(c) shows that PC1 captures the dominant variance, supporting the low-dimensional projection hypothesis. We further evaluate the consistency condition (Eq. 23) using only the PC1 component to approximate the key vectors, i.e., $W_K E_i^T \approx \alpha_i \mathbf{v}$ where $\mathbf{v}$ is the first principal component. Figure 19(a) (green bars) shows the proportion of unseen pairs satisfying this approximated consistency condition. Remarkably, for some heads, the satisfaction rate exceeds that of the full projection without bias (blue bars), suggesting that the low-dimensional subspace structure actively contributes to generalization.

To determine whether this low-dimensionality arises from the low-rankness of $W_K$ itself, we computed the effective rank using singular value decomposition (SVD). Figure 19(d) shows that, with a threshold of $99\%$ cumulative explained variance, the effective rank is at most 6, confirming that $W_K$ is indeed low-rank. Furthermore, we verified that the first right singular vector of $W_K$ aligns with the PC1 vector $\mathbf{v}$ of the key vectors $\{W_K E_i^T\}$, indicating that the low-dimensional structure of the key vectors is inherited from the low-rank structure of $W_K$.

Finally, we examine whether $\rho_a$ and $\rho_b$ exhibit positional asymmetry, i.e., whether $\rho_a > \rho_b$ (or $\rho_a < \rho_b$) holds consistently for all positions with $a < b$. Figure 19(e) shows that serveral heads have a clear horizontal gradient in $\rho$, providing strong support for the theoretical prediction.

In summary, our analysis demonstrates that both Mechanism 1 (bias-induced gradient) and Mechanism 2 (learned subspace alignment) emerge in models trained with RoPE, enabling generalization to unseen object pairs.

### J.3.3. CONCLUSION.

RoPE can enable out-of-distribution generalization through two distinct mechanisms. First, when bias terms are present, the interaction between $b_K$ and position-dependent RoPE rotations creates a positional gradient $\Pi_{\text{bias}}$ analogous to learnable positional embeddings. Second, even without bias terms, generalization can occur if $W_K$ projects token embeddings into a common subspace where RoPE induces a consistent positional asymmetry. Both mechanisms achieve position-dependent attention bias that transfers to unseen object pairs.

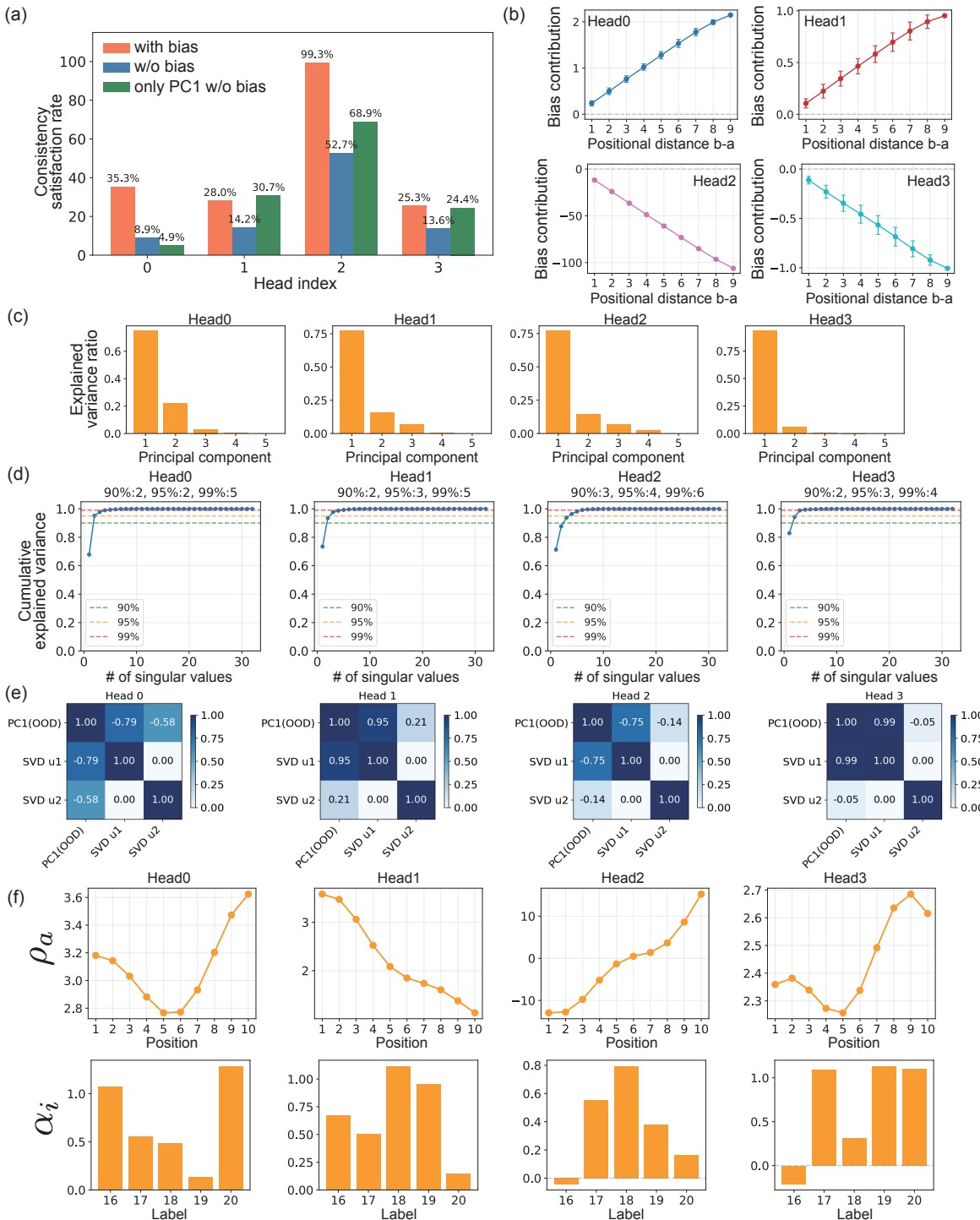

*Figure 19.* Analysis of 1-layer model with RoPE. All analyses are performed on unseen-pair data to investigate the mechanism underlying unseen-pair generalization. The model analyzed in Fig. 18 is investigated. (a) Proportion of unseen object pairs satisfying the consistency condition for each attention head. Orange: the full condition (Eq. 23), Blue: the condition $T_1 T_2 > 0$ with $\Pi_{\text{bias}} = 0$, Green: $T_1 T_2 > 0$ with the approximation in Eq. 24. (b) Bias-induced positional gradient $\Pi_{\text{bias}}$ as a function of positional distance $b - a$. Mean and standard deviation are computed across all position pairs with $a < b$. (c) Explained variance ratio from PCA of the key vectors $\{W_K E_i^T\}$ for labels in the unseen-pair dataset. (d) Cumulative explained variance of the singular values of $W_K$. Effective rank is shown for thresholds of $90\%$, $95\%$, and $99\%$. (e) Cosine similarity between PC1 from (c) and the first two right singular vectors ($\mathbf{u}_1$, $\mathbf{u}_2$) of $W_K$. (f) Position-dependent attention contribution $\rho_a$ as a function of position, along with the corresponding token-specific scalars $\alpha_i$ from Eq. 24.

# K. Extension to 2D images

To examine whether the attention gradient mechanism identified in the 1D setting extends to higher-dimensional inputs, we conduct experiments with 2D images.

## K.1. Setup

We place two objects in $4 \times 4$ images and assign relational text descriptions (e.g., "A is on the left of B") based on their horizontal positions in the 2D image (Fig. 20(a)). For simplicity, we exclude images where two objects share the same horizontal position. The image is flattened into a sequence of 16 tokens, prepended with a CLS token, for input to the transformer. We use the same 1-layer attention-only model architecture and training procedure as in the 1D case.

## K.2. Results

Figure 20(b) shows the generalization accuracy under the same conditions as in the 1D case (Fig. 4). The model generalizes to unseen object pairs with similar accuracy to the 1D setting. For example, with $N_{\text{tot}} = 20, n_2 = 10$, the model achieves 100% on single-object positioning, 100% on seen-pair configurations, and 98% on unseen-pair generalization.

The attention-logit decomposition from the CLS token to all spatial positions (Fig. 20(a) and (c)) reveals horizontal attention gradients arising from positional embeddings, consistent with the 1D case. These results demonstrate that the symmetry-breaking mechanism is not an artifact of the 1D simplification, and persists when the spatial structure is extended to two dimensions. We leave the investigation of more complex 2D spatial relations, such as above/below and their combinations with horizontal relations, for future work.

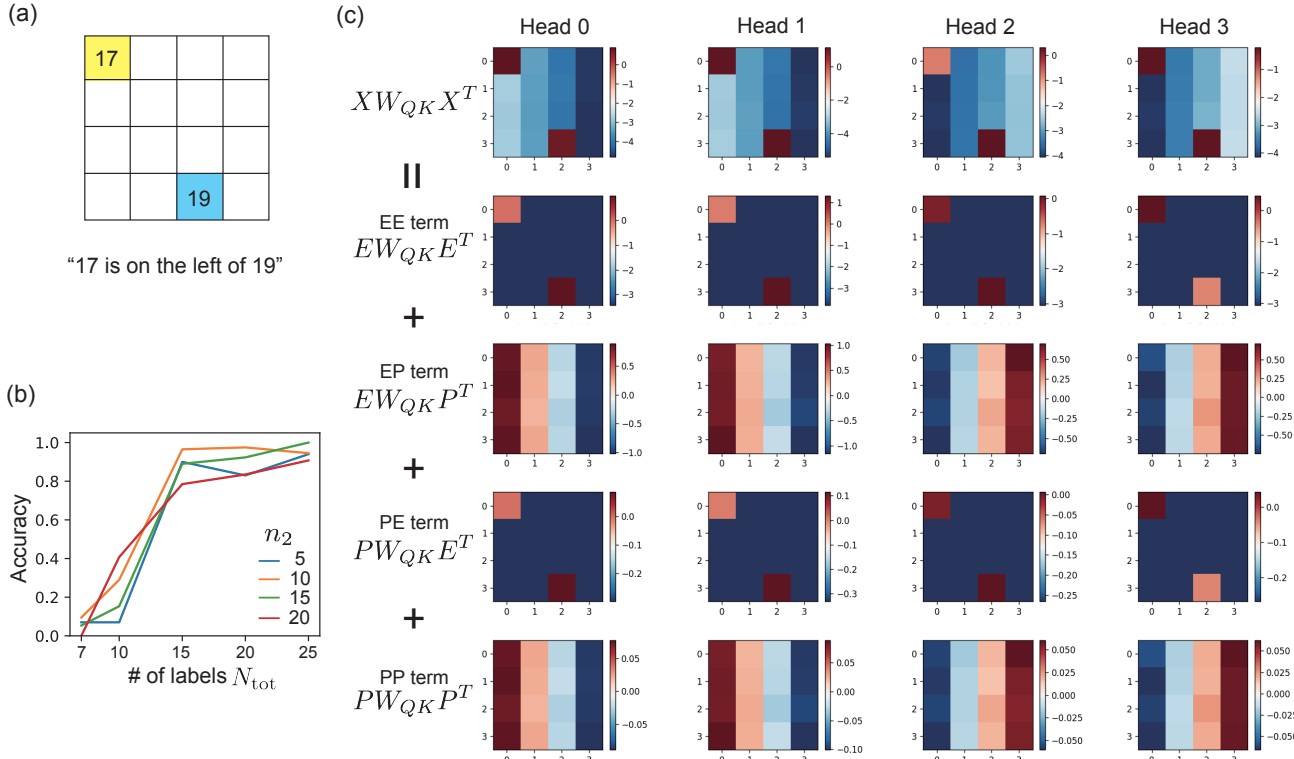

*Figure 20.* Generalization accuracy and attention logit decomposition for 2D images ($4 \times 4$) under the same conditions as in the 1D case. (a) An example of 2D images where object 17 is on the left of object 19. (b) Accuracy for unseen-pair generalization. (c) Attention-logit decomposition (EE, EP, PE, PP terms) from the CLS token to all spatial positions for all heads on the image in (a). $N_{\text{tot}} = 20, n_2 = 10$.

# L. 1 Layer-4 Head Model for Both Vision and Text Encoders Trained with Left and Right Textual Representation

## L.1. Experiment

Figure 21 shows the one-object recognition and generalization to unseen object pairs for models with 1-layer, 4-head vision and text encoders. We find that the models can recognize individual objects (Fig. 21(a)), but fail to generalize to unseen object pairs (zero accuracy as shown in Fig. 21(b)).

To further understand this failure, we relax the criterion for unseen object pair generalization: a prediction is considered correct if the highest similarity is achieved by either the left or right textual representation. Even with this relaxed criterion, the models achieve at most 0.5 accuracy, indicating that they can only recognize which object pair is present in the image as shown in Fig. 21(d), but cannot encode the left–right spatial relationship.

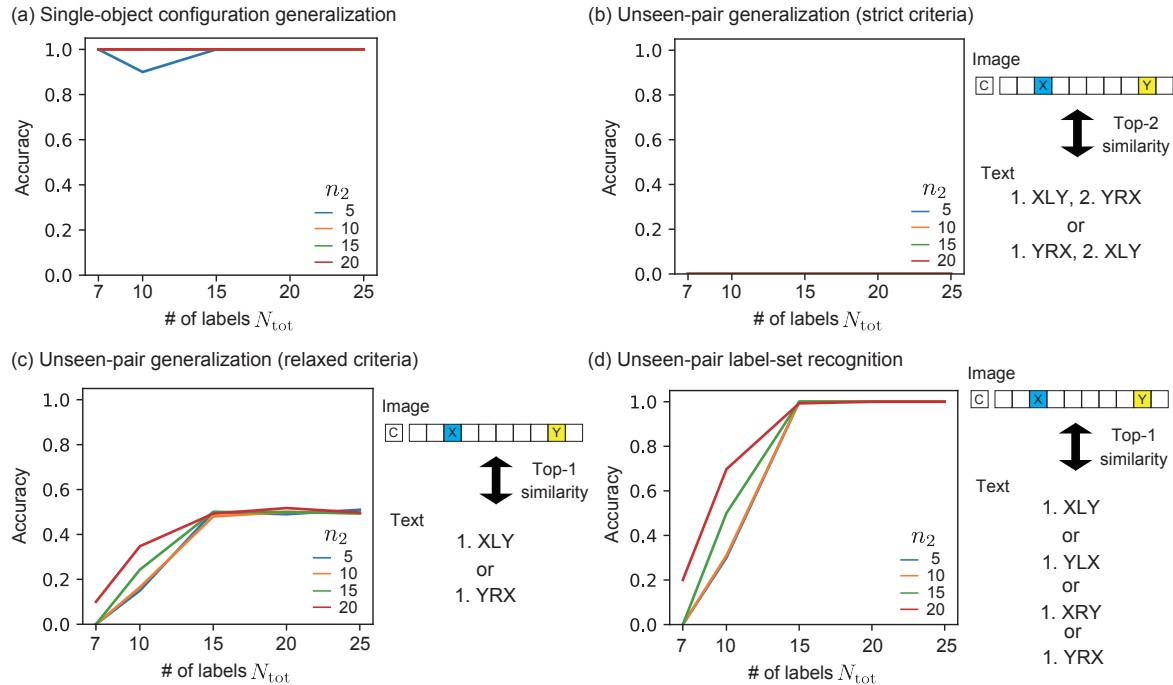

*Figure 21.* Single-object recognition and generalization to unseen object pairs for models with 1-layer, 4-head vision and text encoders. (a) The single-object positional generalization, and (b,c) unseen-pair generalization are shown. In (c), we relax the criteria: a prediction is considered correct if the highest similarity is achieved by either the left or right textual representation. (d) Accuracy for unseen-pair label-set recognition.

## M. Ablation Experiments in the Presence of Both Left and Right Textual Representations

In Sec. 7, we showed that the model generalizes to unseen object pairs when both "left of" and "right of" text representations are used during training (hereafter referred to as the "bidirectional" setting). Here, we verify that the attention gradient mechanism identified in the left-only setting (Sec. 5) remains causally relevant in this bidirectional setting.

We perform the same ablation procedure as in Fig. 5(e): at inference time, we zero specific pre-softmax attention logit terms for all heads and evaluate unseen-pair generalization accuracy. For the left-only setting, accuracy is measured by top-1 retrieval, where the chance level is 50% (the model recognizes the correct object pair but cannot distinguish left from right, as discussed in Sec. 5). For the bidirectional setting, since each image has two correct texts (e.g., "A is on the left of B" and "B is on the right of A"), we measure whether both correct texts occupy the top-2 positions, as defined in Sec. 2.3. Top-1 accuracy (whether any correct text ranks first) is also shown in parentheses for reference. Table 2 summarizes the results across both settings ($n_2 = 10$ for all conditions).

*Table 2.* Ablation of attention logit terms in the left-only and bidirectional settings. Unseen-pair generalization accuracy is reported ($n_2 = 10$). For the bidirectional setting, the top-2 metric is shown, with top-1 accuracy in parentheses.

| Text setting | $N_{tot}$ | Baseline | EP abl. | PP abl. | BP abl. | EP+PP+BP abl. |
|---|---|---|---|---|---|---|
| Left-only | 20 | 82% | 55% | 81% | 81% | 50% |
| Left-only | 25 | 89% | 61% | 88% | 88% | 50% |
| Bidirectional | 20 | 92% (93%) | 64% (68%) | 91% (92%) | 90% (91%) | 44% (50%) |
| Bidirectional | 25 | 82% (91%) | 56% (60%) | 82% (90%) | 82% (89%) | 40% (45%) |

Across all settings, ablating the EP term alone causes the largest accuracy drop (approximately 26–28%), while ablating PP or BP alone causes only a small drop (0–2%). All three terms (EP, PP, BP) exhibit horizontal attention gradients in the bidirectional setting, as in the left-only setting. Ablating all three gradient-bearing terms together (EP+PP+BP) drops accuracy to at or below 50%, confirming that the attention gradient arising from positional embeddings is essential for spatial discrimination regardless of the text representation used.

**Failure Analysis.**   To better understand what the ablated model can and cannot do, we analyzed the failure cases in the bidirectional setting when the EP term or all three terms (EP+PP+BP) are ablated. Among incorrect cases (where the two correct texts do not occupy the top-2), we measure the proportion of cases where the model places the semantically correct object pair but with the wrong spatial ordering in the top-2. For example, when the ground truth is {"A is on the left of B", "B is on the right of A"}, the model instead places {"A is on the right of B", "B is on the left of A"} in the top-2. Table 3 summarizes the results.

*Table 3.* Failure analysis in the bidirectional setting ($n_2 = 10$). Among incorrect cases (where the two correct texts do not occupy the top-2), the proportion of cases where the model selects the correct object pair but with the wrong spatial ordering.

| $N_{tot}$ | Ablation type | Wrong ordering (%) |
|---|---|---|
| 20 | EP abl. | 75% |
| 20 | EP+PP+BP abl. | 73% |
| 25 | EP abl. | 66% |
| 25 | EP+PP+BP abl. | 63% |

Across all conditions, 63–75% of the failures involve the correct object pair with the wrong spatial ordering. This suggests that the ablated model retains some ability to identify the relevant objects and select the corresponding semantic text pair, but tends to assign the wrong spatial ordering. Overall, these ablation experiments show that the attention gradient, driven primarily by the EP term with additional support from the BP and PP terms, is the causal mechanism for left–right spatial discrimination also in the bidirectional setting.

# N. Extension to 3-Object Images

To examine whether the attention gradient mechanism extends beyond the 2-object setting, we conduct experiments with 3-object images. In a 3-object scene, each object participates in multiple spatial relations simultaneously (e.g., object B can be right of A and left of C), making a simple two-role assignment ("left object" and "right object") insufficient.

## N.1. Evaluation of 2-Object Trained Models on 3-Object Scenes

We first evaluate whether models trained on single- and 2-object images (i.e., the models from Fig. 4) can transfer to 3-object spatial discrimination. We evaluate on 3-object scenes composed of object labels that do not appear in the 2-object training data (i.e., OOD generalization to unseen labels), where three pairwise relations hold simultaneously (A left of B, A left of C, B left of C).

We use the following metrics for 3-object evaluation: "3obj top-1" measures whether any correct text ranks first; "3obj top-3 mean" is the average number of correct texts in the top-3; and "3obj top-3 all" measures whether all three correct texts appear in the top-3.

The best performing model ($N_{\text{tot}} = 20, n_2 = 15$ in Fig. 4(a)) achieves approximately 83% on "3obj top-1", substantially above chance. However, it recovers on average about 1.8 out of 3 correct relations ("3obj top-3 mean"), and achieves at most 15% on "3obj top-3 all". These results show that the 2-object trained model partially transfers to 3-object scenes, but does not fully generalize. In the next section, we address this by training directly on 3-object images.

## N.2. Direct Training with 3-Object Images

We next train models on single-, 2-, and 3-object images jointly via contrastive learning. For 3-object images of the training dataset, we used only the labels that appear in the 2-object training dataset. Since each 3-object image has three valid text descriptions, we randomly select one as the text pair in the training dataset and mask the remaining two from the negative set in each training batch to mitigate the possible effects of false negatives in the contrastive objective. For each triplet of objects in the 3-object images of the training dataset, two positional configurations are sampled uniformly at random. We evaluate on 3-object images composed of object labels that do not appear in any 2-object or 3-object training data (i.e., OOD generalization to unseen labels) using the above metrics "3obj top-1", "3obj top-3 mean" and "3obj top-3 all". "2obj OOD" is the unseen-pair generalization accuracy for 2-object images.

All models hereafter use attention-only encoders (without MLP or LayerNorm). With a single-layer vision encoder, the models achieve 35% (trained with 1-layer text encoder) and 56% (trained with 2-layer text encoder) on "3obj top-3 all", both significantly above chance, demonstrating that generalization to 3-object scenes emerges ($N_{\text{tot}} = 20, n_2 = 10$). Further parameter tuning may improve performance for both configurations. With 2-layer vision and 2-layer text encoders, performance improves further. For example, a model with strong generalization achieves 98% on "3obj top-1" and 82% on "3obj top-3 all" ($N_{\text{tot}} = 20, n_2 = 10$), as summarized in Table 4.

*Table 4.* Ablation of attention logit terms in the first layer of the 2-layer vision encoder trained on single-, 2-, and 3-object images jointly.

| Ablation | 2obj OOD | 3obj top-1 | 3obj top-3 mean | 3obj top-3 all |
|---|---|---|---|---|
| Baseline | 95% | 98% | 2.82 | 82% |
| EP abl. | 65% | 69% | 1.98 | 28% |
| PP abl. | 95% | 97% | 2.81 | 81% |
| BP abl. | 95% | 98% | 2.81 | 82% |
| EP+PP+BP abl. | 52% | 58% | 1.68 | 17% |

In both single-layer and 2-layer vision encoder cases, we found that horizontal attention gradients emerge in the EP, PP, and BP terms in the first layer of the vision encoder, consistent with the 2-object case (Fig. 22). Ablating the EP term alone causes the largest drop (e.g., 82% → 28% on 3obj top-3 all), while ablating PP or BP alone has minimal effect. Ablating all three gradient-bearing terms together drops accuracy further (e.g., 82% → 17%). These results indicate that the attention gradient mechanism, driven primarily by the EP term, is causally relevant in the 3-object setting. In the 2-layer model, the attention gradient is present and causally important in the first layer, but other mechanisms in the second layer may also contribute to the improved performance. Disentangling the contributions of each layer and exploring whether 1-layer models

can achieve comparable performance with alternative training strategies are important directions for future work.

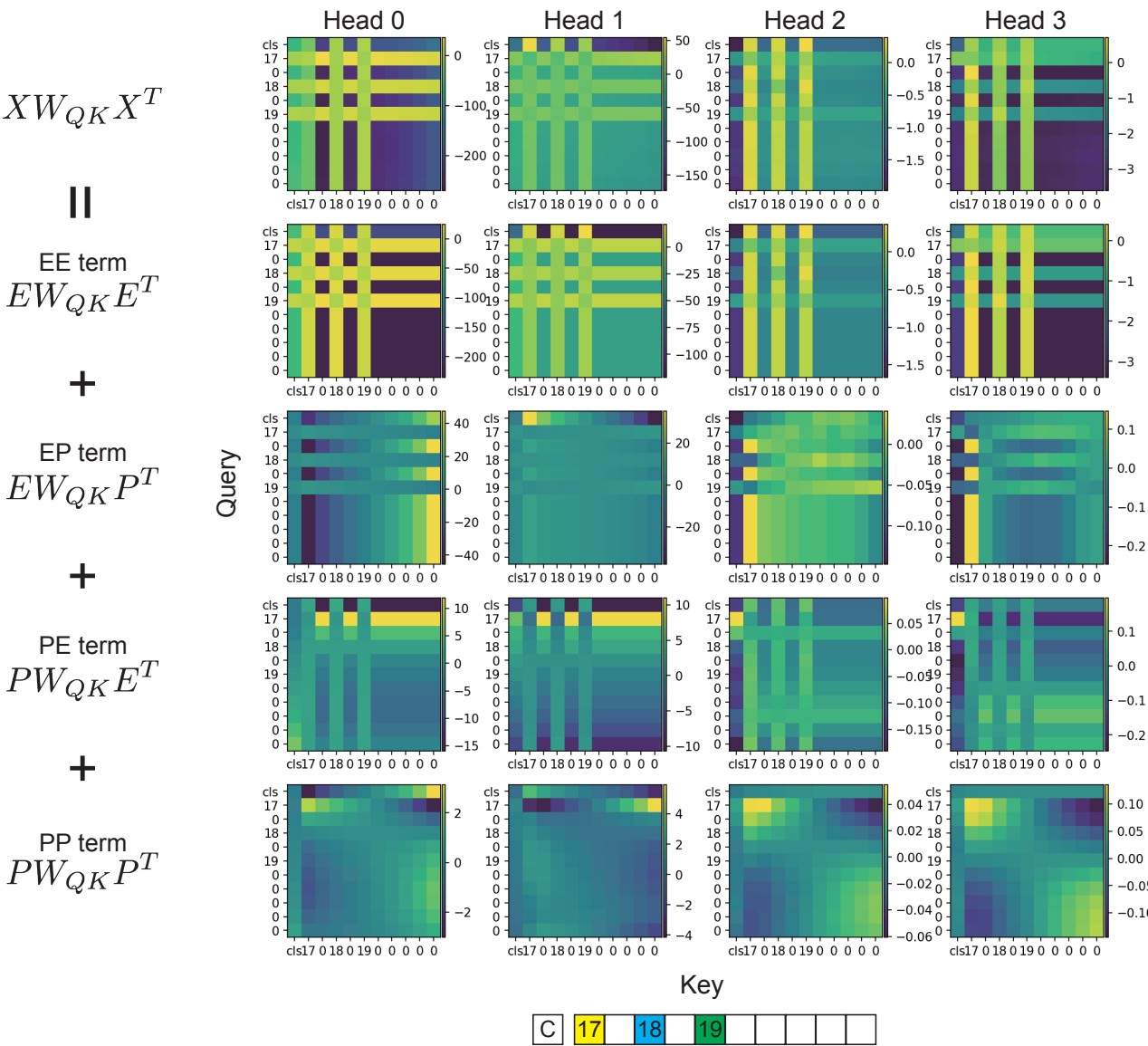

*Figure 22.* Attention patterns observed in the first Transformer layer of a generalizing model with 2-layer vision encoder trained on 3-object images. Positional-token embedding decomposition of pre-softmax logits for all the heads is shown. The model is the one discussed in Sec. N.2 ($N_{\text{tot}} = 20, n_1 = 5, n_2 = 10, N_{\text{val}} = 5$.)

# O. Extension to Autoregressive VLM

To examine whether the attention gradient mechanism is specific to contrastive training, we train an autoregressive VLM on the same toy setting. We use randomly initialized, frozen token embeddings for each object label as a minimal vision encoder, deliberately excluding positional embeddings so that spatial information must come from the decoder. The vision tokens are mapped to the text space via an MLP and fed into the decoder in spatial order. The decoder's own learnable positional embeddings provide the spatial information. We train a 2-layer attention-only Transformer decoder with next-token prediction: the input is [10 vision tokens (one per pixel)] [BOS] ["A"] ["is left of"] ["B"], and the target is ["A"] ["is left of"] ["B"] [EOT]. The vision tokens attend bidirectionally to each other, while text tokens use causal attention. The model generalizes to unseen object pairs: for example, at $N_{\text{tot}} = 40, n_2 = 10$, it achieves 100% accuracy on single-object positional and seen-pair configuration generalization, and 94% on unseen-pair generalization, where accuracy is measured by free generation of the complete relational text. We note that a 1-layer decoder does not generalize in this setting, though we cannot rule out that further hyperparameter tuning might resolve this.

As shown in Fig. 23, our attention-logit decomposition of the first layer reveals that the EP and PP terms exhibit a clear attention gradient from text token queries over the vision token keys, preferentially attending to one side of the image sequence, consistent with the CLIP case (Fig. 5). This demonstrates that the symmetry-breaking mechanism is not specific to contrastive training but emerges also in autoregressive training where the decoder is responsible for spatial reasoning.

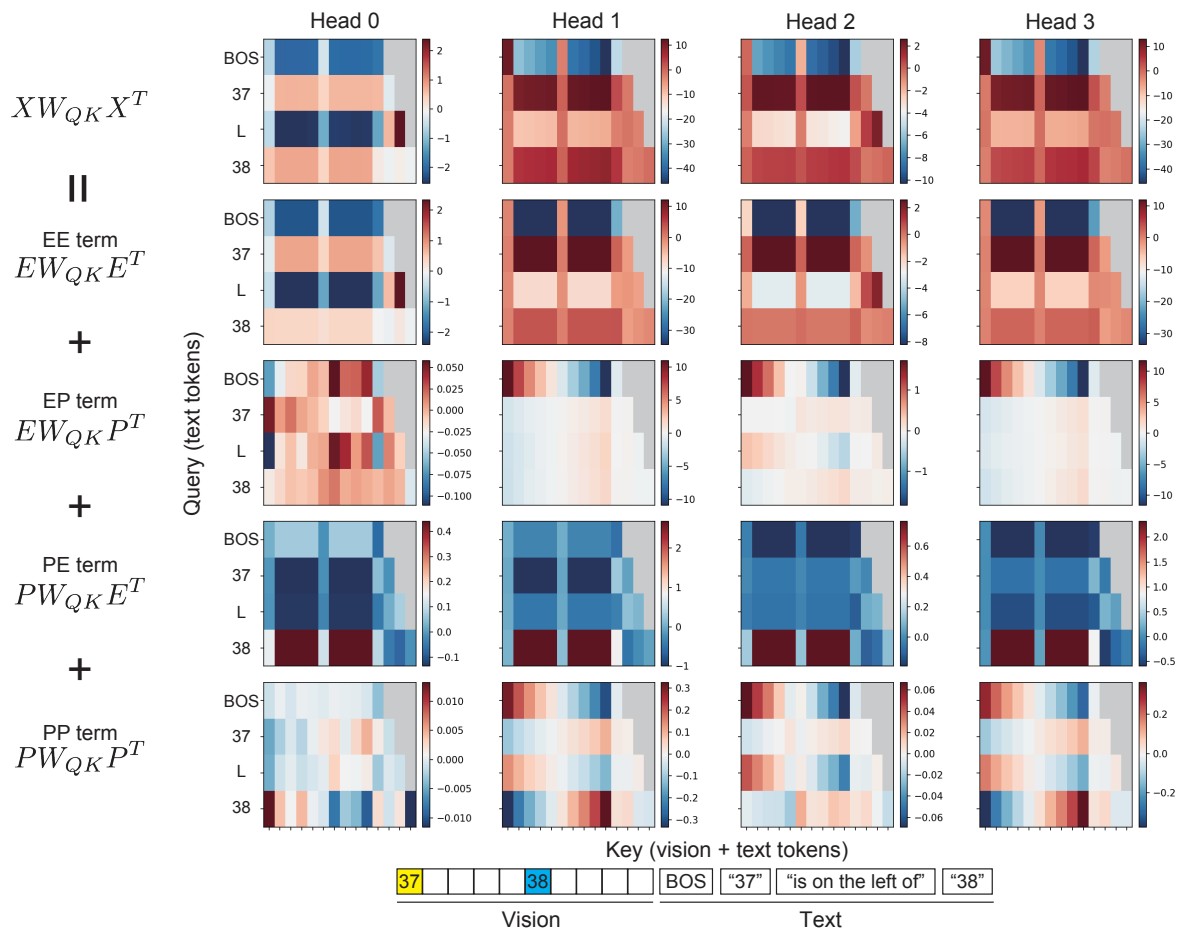

*Figure 23.* Autoregressive VLM: attention-logit decomposition. Attention-logit decomposition (EP, PE, PP, EE terms) from text tokens to preceding tokens for all heads, shown for a representative unseen-pair image (OOD) with objects 37 and 38. Gray regions indicate the causal mask, where attention is not permitted. The model is trained with $N_{\text{tot}} = 40, n_2 = 10$, achieving 94% unseen-pair generalization accuracy (100% on single-object positioning and seen-pair configurations), measured by free generation of the complete relational text.

## P. Effect of Number of Heads on Generalization

In Fig. 24, we vary the weight decay regularization and model capacity to investigate whether a single-head, single-layer vision Transformer can discriminate left–right relations. We keep the text encoder as 4 head, single layer Transformer. Across all hyperparameter configurations tested, the accuracy of unseen-pair generalization remains around or below 0.5.

We also test whether high accuracy can be achieved by pruning heads from a generalizing four-head model and continuing training, hypothesizing that this initialization might facilitate learning. Figure 25 shows that models retaining multiple heads achieve accuracy close to that of the original four-head model, whereas accuracy remains near 0.5 for the single-head model. Among the two-head models that achieve high accuracy (i.e., those retaining heads $[0, 2]$, $[1, 2]$, or $[2, 3]$), we observe the horizontal gradient in the EP term of the pre-softmax logits, consistent with our findings in the main text (Fig. 25(b)). Interestingly, in each of these cases, the two retained heads are both left-biased and right-biased, respectively, suggesting that the presence of complementary specialist heads enhances left–right discrimination.

These results suggest that multi-head attention may be necessary for left–right discrimination in our setting, though we cannot rule out the possibility that a single-head, single-layer vision Transformer could succeed under different conditions. Further experimental and theoretical investigation is required to better understand the role of multi-head attention in generalization.

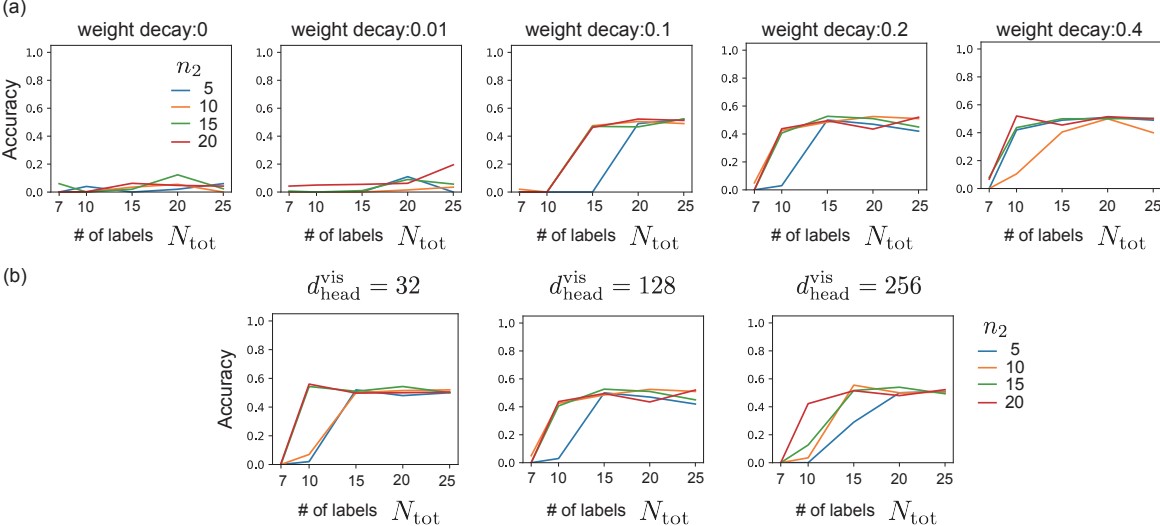

*Figure 24.* Performance of models with single-head, single-layer vision encoder. Accuracy for unseen-pair generalization is shown. (a) Effect of weight decay regularization. The experiments are performed for the model with $M_B^{\text{vis}} = M_{\text{rep}}^{\text{vis}} = 1$, $M_h^{\text{vis}} = 1$, $d_{\text{head}}^{\text{vis}} = d_{\text{model}}^{\text{vis}} = 128$, $M_B^{\text{txt}} = M_{\text{rep}}^{\text{vis}} = 1$, $M_h^{\text{vis}} = 4$, $d_{\text{head}}^{\text{txt}} = 32$, $d_{\text{model}}^{\text{txt}} = 128$, by keeping the other parameters. (b) Effect of the Transformer head dimensionality $d_{\text{head}}^{\text{vis}}$ of the vision encoder. The other parameters are the same as those used in weight decay $w = 0.2$ case in (b).

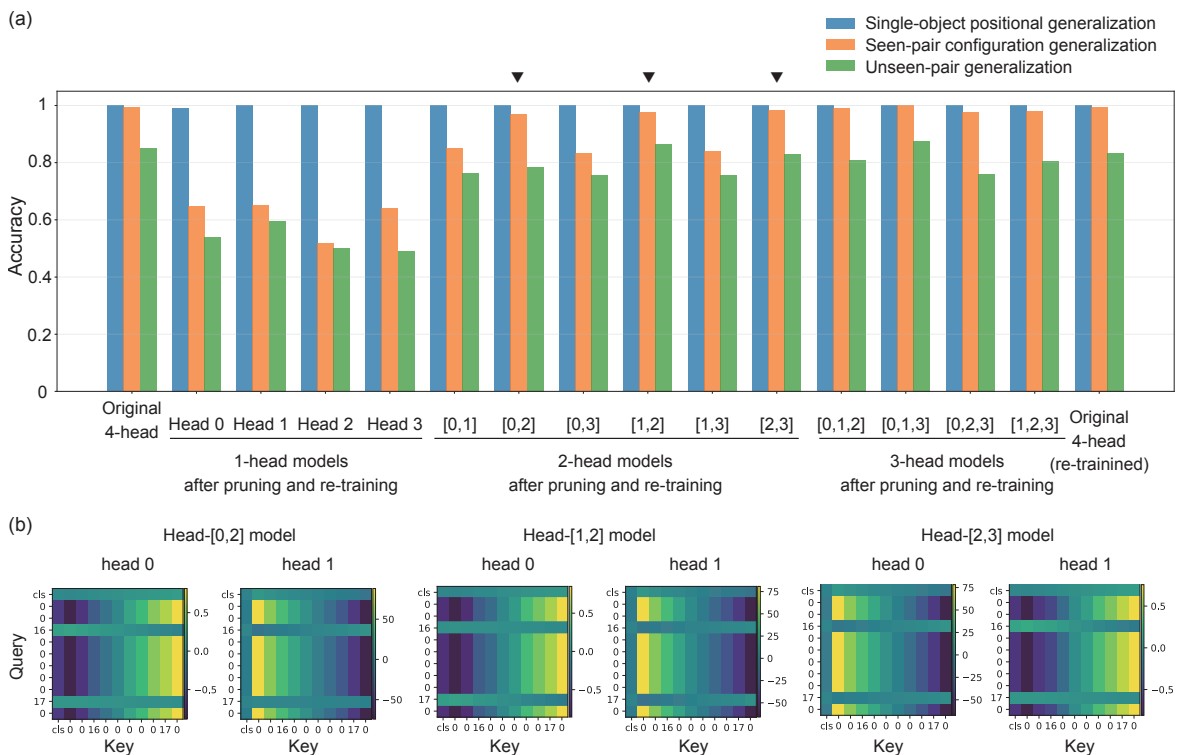

*Figure 25.* Performance of models with pruned heads after retraining. Models are obtained by pruning heads from a trained four-head, single-layer model and retraining for an additional 10,000 epochs. (a) Performance of original and pruned models for all possible head pruning configurations. Triangles indicate two-head models that achieve high accuracy across all three generalization types. (b) Contribution of the EP term to pre-softmax logits for two-head models with high generalization accuracy.

## Q. Attention Pattern Observed in Generalizing Model with 2-Layer Vision Encoder

Here, we demonstrate the emergence of attention gradients in a generalizing model with a 2-layer architecture (accuracy: 1.0 for single-object positional, 1.0 for seen-pair configuration, and 0.86 for unseen-pair generalization). We extend the reduced 1-layer model from Sec. 4 by replacing the vision encoder with a 2-layer variant ($M_B = 2, M_{rep} = 1, M_h = 4$) while retaining the 1-layer text encoder.

Following the same approach as for the 1-layer model, we decompose the contributions to the pre-softmax logit into several terms for the first layer of the vision encoder. As shown in Fig. 26, we observe horizontal attention gradients in the contributing terms of the first layer, similar to those in the 1-layer model. Nevertheless, the second layer may introduce additional mechanisms that increase the overall complexity of the attention dynamics. A thorough investigation of how the two layers interact, including disentangling the contributions of each layer, is an important direction for future work.

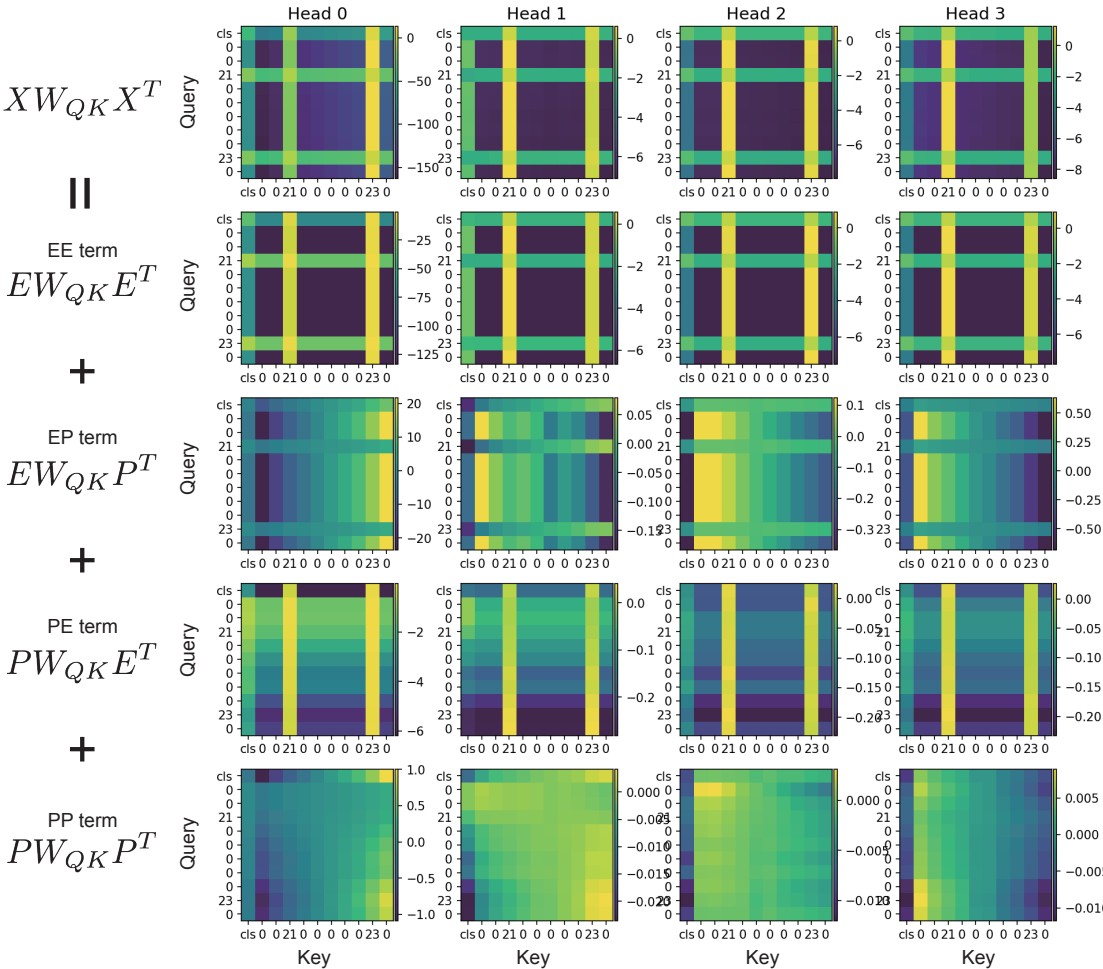

*Figure 26.* Attention pattern observed in the first Transformer layer of a generalizing model with 2-layer vision encoder. Positional-token embedding decomposition of pre-softmax logit for all the heads of generalizing model. The parameter for the dataset generation is $N_{\text{tot}} = 25, n_1 = 5, n_2 = 15, N_{\text{val}} = 5$.

## R. Approximate Attention Decomposition with LayerNorm

In the main text, we analyze the attention mechanism in attention-only models (without MLPs or LayerNorm), where the clean decomposition into EE, EP, PE, and PP terms is exact. Here, we take a preliminary step toward analyzing non-linear models by proposing an approximate decomposition of the LayerNorm layer. We focus on the first layer, where the input is the sum of token and positional embeddings $X = E + P$, allowing us to separate their contributions. Extending the decomposition to deeper layers, where the input is the residual stream accumulating outputs from all previous layers, requires further approximation techniques and is left for future work.

### R.1. Decomposition

In a standard Transformer with pre-LayerNorm, the input to the first attention layer is $\text{LN}(X)$ rather than $X$ directly, where $X = E + P$. LayerNorm computes, for each token position:

$$\text{LN}(X) = \gamma \cdot \frac{X - \text{mean}(X)}{\text{std}(X)} + \beta, \tag{32}$$

where $\text{mean}(\cdot)$ and $\text{std}(\cdot)$ are computed across the embedding dimension, $\cdot$ denotes element-wise multiplication and $\gamma, \beta \in \mathbb{R}^d$ are learnable scale and shift parameters.

Since $X = E + P$, the centered input decomposes as:

$$X - \text{mean}(X) = (E - \text{mean}(E)) + (P - \text{mean}(P)) = \hat{E} + \hat{P}, \tag{33}$$

where $\hat{E}$ and $\hat{P}$ are the mean-centered token and positional embeddings respectively. Substituting into Eq. 32, we obtain:

$$\text{LN}(X) = \frac{\gamma\hat{E}}{\text{std}(X)} + \frac{\gamma\hat{P}}{\text{std}(X)} + \beta. \tag{34}$$

Note that the coupling between $E$ and $P$ remains in $\text{std}(X) = \text{std}(\hat{E} + \hat{P})$, so this decomposition is approximate. Nevertheless, it separates the LayerNorm output into three interpretable components: an $\hat{E}$-term (token embedding contribution), a $\hat{P}$-term (positional embedding contribution), and a $\beta$-term (bias contribution).

### R.2. 9-Term Expansion of Attention Logits

In the attention-only model, the weight-related component $XW_{QK}X^T$ of the pre-softmax attention logits (Eq. 1) yields $2 \times 2 = 4$ terms when decomposed into $E$ and $P$ contributions (Eq. 2). With LayerNorm, the corresponding weight-related component becomes $\text{LN}(X)W_{QK}\text{LN}(X)^T$. Since each $\text{LN}(X)$ has three components ($\hat{E}$-term, $\hat{P}$-term, $\beta$-term), this yields $3 \times 3 = 9$ terms:

$$\begin{aligned}
\text{LN}(X)W_{QK}\text{LN}(X)^T = {} & \frac{\gamma\hat{E}}{\text{std}(X)}W_{QK}\frac{(\gamma\hat{E})^T}{\text{std}(X)} + \frac{\gamma\hat{E}}{\text{std}(X)}W_{QK}\frac{(\gamma\hat{P})^T}{\text{std}(X)} + \frac{\gamma\hat{E}}{\text{std}(X)}W_{QK}\beta^T \\
& + \frac{\gamma\hat{P}}{\text{std}(X)}W_{QK}\frac{(\gamma\hat{E})^T}{\text{std}(X)} + \frac{\gamma\hat{P}}{\text{std}(X)}W_{QK}\frac{(\gamma\hat{P})^T}{\text{std}(X)} + \frac{\gamma\hat{P}}{\text{std}(X)}W_{QK}\beta^T \\
& + \beta W_{QK}\frac{(\gamma\hat{E})^T}{\text{std}(X)} + \beta W_{QK}\frac{(\gamma\hat{P})^T}{\text{std}(X)} + \beta W_{QK}\beta^T.
\end{aligned} \tag{35}$$

We denote these nine terms as $\hat{E}\hat{E}$, $\hat{E}\hat{P}$, $\hat{E}\beta$, $\hat{P}\hat{E}$, $\hat{P}\hat{P}$, $\hat{P}\beta$, $\beta\hat{E}$, $\beta\hat{P}$, and $\beta\beta$, respectively. The original 4-term decomposition (EE, EP, PE, PP) in the attention-only model corresponds to the $\hat{E}\hat{E}$, $\hat{E}\hat{P}$, $\hat{P}\hat{E}$, and $\hat{P}\hat{P}$ terms here, with the additional $\beta$-related terms arising from the LayerNorm bias parameter.

### R.3. Results

We investigated the contribution of the 9 terms in the first layer of the multi-layer non-linear model ($N_{\text{tot}} = 20, n_2 = 20$ in Fig. 3). As shown in Fig. 27, we found that the $\hat{E}\hat{P}$, $\hat{P}\hat{P}$ and $\beta\hat{P}$ terms exhibit horizontal attention gradients consistent with those identified in the attention-only model (without LayerNorm or MLP). These results suggest that the symmetry-breaking mechanism persists in the first layer in the presence of non-linear components, although additional mechanisms are likely also contributing. A thorough validation of this approximate decomposition and its implications, including extending the analysis to deeper layers to the non-linear setting, is an important direction for future work.

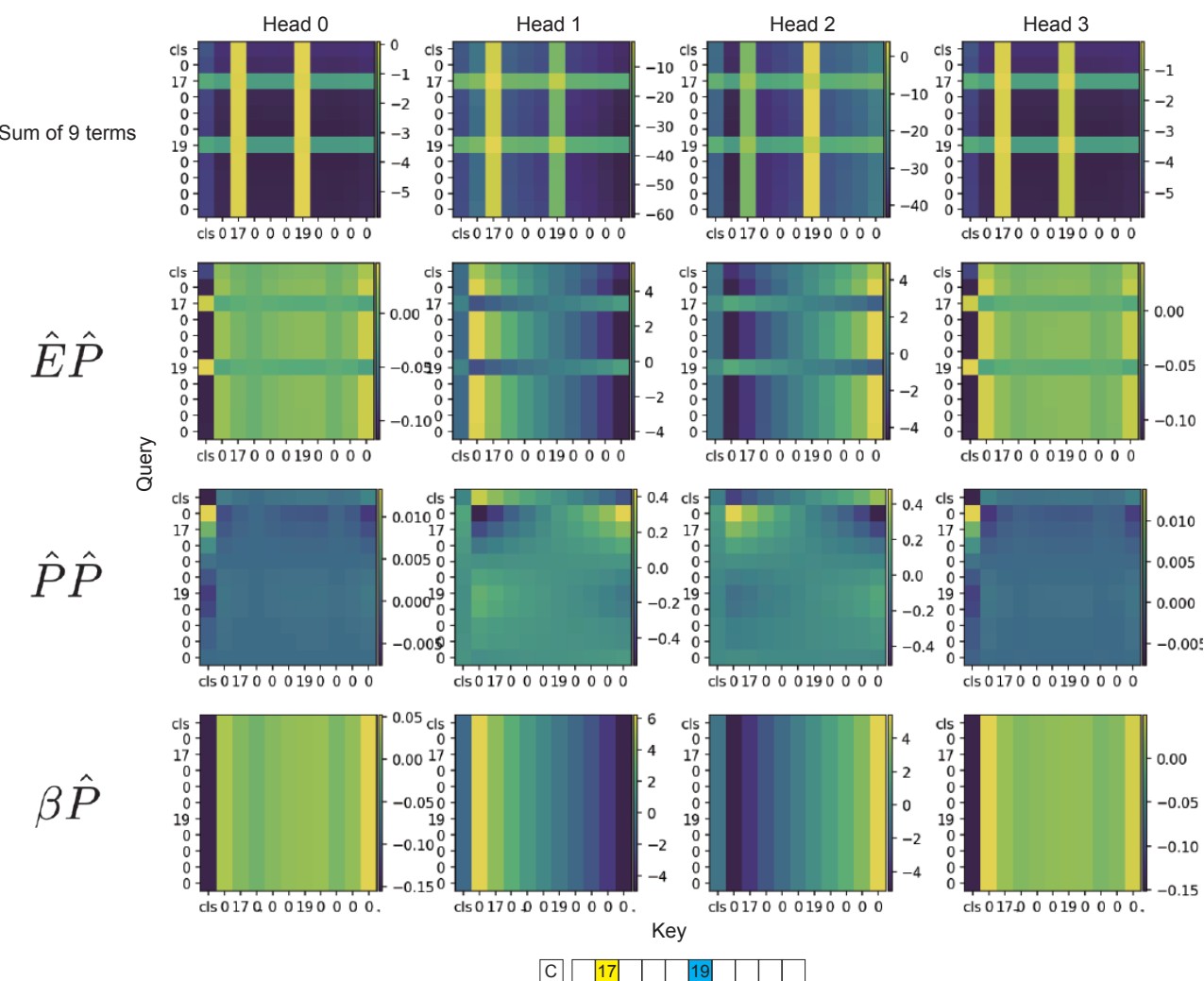

*Figure 27.* Approximate attention decomposition with LayerNorm. The sum of 9 terms, $\hat{E}\hat{P}$, $\hat{P}\hat{P}$ and $\beta\hat{P}$ terms are shown for a 1D image with unseen-pair of objects 17 and 19. The model is trained with the condition $N_{\text{tot}} = 20, n_2 = 20$ in Fig. 3.

