# OpenReview forum: "Left–Right Symmetry Breaking in CLIP-Style Vision-Language Models Trained on Synthetic Spatial-Relation Data"
_ICML.cc/2026/Conference — ICML 2026 regular_

### Official Review · Reviewer_S34a · 2026-03-11

**Soundness:** 2
**Presentation:** 3
**Significance:** 2
**Originality:** 2
**Overall Recommendation:** 3
**Confidence:** 3

**Summary:**

This paper investigates how vision and language models acquire spatial reasoning capabilities by training simplified Transformer architectures on a synthetic sequence dataset. The authors demonstrate that contrastive training enables these models to learn basic spatial relations and successfully generalize to novel pairs of objects. Through a mathematical decomposition of the attention weights, the study reveals that the interaction between positional and token embeddings generates a distinct horizontal attention gradient. This gradient breaks the spatial symmetry of the inputs and serves as the primary mechanism for the vision encoder to discriminate relative object positions.

**Compliance With Llm Reviewing Policy:**

Affirmed.

**Final Justification:**

The authors provided rigorous mathematical decompositions and new 2D experiments in the rebuttal, the fundamental gap between their simplified toy models and real-world VLM remains large. The semantic complexity, network depth, and rich visual features of actual CLIP models are vastly different from the highly constrained 1D and 2D synthetic environments evaluated here. Without empirical evidence showing that this specific shallow attention mechanism actually dictates spatial reasoning in production-level models, therefore I maintain the score.

**Key Questions For Authors:**

1. While Section 3 demonstrates that a full architecture (with MLPs and LayerNorm) successfully generalizes, the mathematical decomposition (the $EW_{QK}P^T$ gradient) is exclusively performed on an ablated, attention-only model in Section 4. Given that high-dimensional MLPs excel at complex non-linear feature binding and often learn shortcuts, what concrete empirical proof (e.g., logit attribution on the full model) confirms that this specific horizontal gradient remains the actual causal driver of spatial reasoning, rather than being completely bypassed or overridden by uninterpretable MLP transformations?
2. The authors motivate this work with safety-critical applications and claim to provide 'practical guidance.' However, Section 7 reveals that introducing simple bidirectional synonyms ('left of' / 'right of') causes a capacity collapse in the 1-layer model. While adding a second layer restores accuracy, the authors explicitly admit its internal dynamics 'remain unclear due to the complexity' (Line 412). If mechanistic transparency breaks down entirely upon encountering the most basic linguistic variation, isn't the claim of providing mechanistic insight for real-world, highly complex deployments fundamentally overstated?
3. The mechanistic findings rely almost exclusively on a single-layer setup. As acknowledged in Appendix L, extending the vision encoder to just 2 layers results in 'more complex gradient patterns' that preclude straightforward analysis. Given this profound limitation, how do the authors mitigate the theoretical concern that the observed horizontal gradient might merely be a shallow architectural artifact, a mathematical 'shortcut' forced by the severe lack of network depth, rather than a fundamental mechanistic building block that actually persists in modern, multi-layered VLMs?

**Limitations:**

The primary limitation of this work lies in its heavy reliance on a highly simplified 1D toy dataset and a single-layer, attention-only architecture, which inherently bypasses the complex non-linear dynamics of MLPs and layer-to-layer interactions found in standard deep VLMs. Furthermore, the identified horizontal gradient mechanism exhibits significant fragility, as evidenced by the model's capacity collapse when simply exposed to synonymous bidirectional text variations. Consequently, without empirical validation on deeper networks and continuous 2D spatial environments, it remains fundamentally unverified whether these shallow mechanistic findings can robustly scale to practical, real-world applications.

**Strengths And Weaknesses:**

**Strength:**
1. **Carefully constructed dataset:**
The authors introduce a meticulously designed, synthetic 1D dataset that effectively isolates spatial relationship tasks from complex visual noise. This highly controlled setup successfully creates an environment where the key decision-making processes of the attention mechanism can be cleanly exposed and rigorously analyzed without compounding variables.
2. **Deconstruction of the attention mechanism:**
The paper provides a rigorous mathematical deconstruction of the pre-softmax attention logits, breaking them down into distinct structural components such as the interaction between token and positional embeddings (the EP term). This mechanistic approach offers a transparent and highly interpretable view into how the model fundamentally encodes relational information.
3. **Breaking spatial left-right symmetry:**
Building upon the detailed attention decomposition, the paper elegantly demonstrates how left-right spatial symmetry is broken through the emergence of a horizontal attention gradient. This specific finding provides a compelling and intuitive explanation for how the network acquires directional awareness and achieves zero-shot generalization to unseen object pairs.

**Weaknesses:**
1. **Oversimplification of the attention mechanism:**
The methodology relies on an extreme simplification of the Transformer architecture, notably removing critical non-linear components such as MLPs and LayerNorm. While this abstraction facilitates clear mathematical interpretability, it risks oversimplifying the highly complex attention dynamics that drive standard, real-world vision-language models.
2. **The problem of capacity collapse:**
The experiments reveal a notable capacity collapse when the 1-layer model is exposed to semantically equivalent but syntactically bidirectional text prompts (e.g., both 'left of' and 'right of'). This vulnerability suggests that the identified positional gradient mechanism might be too brittle to handle realistic linguistic diversity, raising concerns about its robustness beyond constrained toy environments.
3. **Bypassing deep networks:**
The study largely bypasses the intricate non-linear transformations and hierarchical feature binding inherent to multi-layered deep networks by focusing almost entirely on a single-layer setup. Since deep layers are essential for complex relational reasoning in modern VLMs, this omission leaves a significant gap in demonstrating whether the proposed shallow mechanism truly scales to deeper, state-of-the-art architectures.

---

> ### Author Rebuttal · Authors · 2026-03-31
>
> We thank the reviewer for the detailed engagement and for recognizing the strengths of our "meticulously designed" dataset, "rigorous mathematical deconstruction" of attention, and the "elegant" demonstration of how the network "acquires directional awareness and achieves zero-shot generalization." We address the concerns and questions below.
>
> ## W1/Q1: Non-linearity
>
> We agree that removing non-linear components simplifies real-world VLMs. We view our attention-only Transformer analysis as a first-order approximation, isolating the attention mechanism in its simplest form as a necessary starting point. As the reviewer notes, Sec. 3 shows that the full architecture generalizes, and Sec. 4 shows the ablated model also generalizes with similar accuracy, suggesting the attention mechanism alone is sufficient for spatial discrimination, though not necessarily the only mechanism at work. We believe that developing methods to analyze non-linearity in Transformers is a challenging but important direction, and we have begun exploring this as described below.
>
> Encouraged by the reviewer's comment, ***we tackled the non-linearity of LayerNorm in the first layer.*** Briefly, for the input $X = E + P$, we can derive $LN(X) = \gamma \hat{E}/\text{std}(X) + \gamma \hat{P}/\text{std}(X) + \beta$, where $\hat{E}$ and $\hat{P}$ are the mean-centered token and positional embeddings respectively. The coupling between $E$ and $P$ remains in $\text{std}(X)$, so this is an approximate decomposition. The three components ($\hat{E}$-term, $\hat{P}$-term, $\beta$-term) yield 3 × 3 = 9 terms in the attention logits. ***We investigated the contribution of these 9 terms in the first layer of the multi-layer full model in Fig. 2 and found that the $\hat{E}\hat{P}, \beta \hat{P}$ and $\hat{P}\hat{P}$ terms exhibit horizontal attention gradients consistent with those identified in attention-only models (w/o LN and MLP).*** This suggests that the symmetry-breaking mechanism persists in the presence of non-linear components, although additional mechanisms are likely also contributing. A thorough validation of this approximate decomposition and its implications is an important future direction. We plan to include this preliminary analysis in the revision.
>
> ## W2/Q2: Capacity collapse
>
> We appreciate this concern but believe the capacity collapse is informative rather than a limitation. The collapse in the 1-layer model with bidirectional prompts is a text encoder capacity issue, not a failure of the spatial mechanism: a 1-layer text encoder cannot represent two inverse relational terms distinctly, and adding a second layer resolves this. This reveals minimum architectural requirements for the mechanism, which is precisely the kind of insight a mechanistic study should provide. We expect that more complex spatial relations will require deeper architectures, and understanding how non-linear components contribute is an important future direction, as discussed in W1/Q1.
>
> We acknowledge that the claim of "practical guidance" is overstated. We will revise, e.g.: "...providing mechanistic understanding that may inform the design of models that reliably encode spatial relations, for instance, by highlighting the role of positional embeddings and the minimum architectural depth required for symmetry breaking."
>
> ## W3/Q3: Multi-layer
>
> We agree that understanding mechanisms in multi-layer models is essential. At the same time, the mechanism emerging in the 1-layer model is itself non-trivial. The discovery that positional-token interactions induce a horizontal gradient breaking left-right symmetry required careful decomposition, and would have been harder to identify in a multi-layer model where multiple mechanisms obscure each other. We view this as a necessary first step. In App. L, we show that the attention gradient emerges in the first layer of a 2-layer vision Transformer, though we acknowledge that additional mechanisms in deeper layers may also contribute. We note that the original phrasing in App. L ("resulting in more complex gradient patterns") was imprecise — the attention gradient is present in the first layer (Fig. 23), and the complexity arises from interactions between layers. We will revise this phrasing. We plan to extend the analysis to multi-layer models for more complex relational understanding as future work.
>
> ## Limitations
>
> We thank the reviewer for this thorough summary and ***have taken concrete steps during the rebuttal period: extending to 2D images (4x4) where the attention gradient emerges (see reply to Reviewers LgEH and LdK1), proposing an approximate LayerNorm decomposition (see W1/Q1), and training an autoregressive VLM with a 2-layer attention-only decoder that also generalizes (e.g. 94% unseen-pair generalization accuracy) and shows the attention gradient in the vision tokens as observed in our CLIP case (see reply to Reviewer LdK1).*** We view these as encouraging first steps toward more realistic settings.

---

> > ### Author Rebuttal · Reviewer_S34a · 2026-04-02
> >
> > I thank the authors for their response. While the authors show the EP gradient is present in the vision encoder under the bidirectional setting (Fig. 8d), the corresponding ablation of the EP term, as performed in Fig. 5(e) for the left-only setting, was not conducted. Could the authors perform this ablation in the bidirectional setting to confirm that the EP gradient retains the same causal role?

---

> > > ### Author Response · Authors · 2026-04-03
> > >
> > > We thank the reviewer for this focused and constructive follow-up. We performed the EP ablation of the vision encoder in the bidirectional setting, where both "left of" and "right of" text prompts are used during training (Section 7, Fig.8). The results confirm that the EP term retains its causal role. We summarize the ablation results (unseen-pair generalization accuracy) across settings in the table below:
> > > | Text setting | $N_{\text{tot}}$ | $n_2$ | Baseline | EP abl. | PP abl. | BP abl. | EP+PP+BP abl. |
> > > |---|---|---|---|---|---|---|---|
> > > | Left-only (Fig.5) | 20 | 10 | 82% | 55% | 81% | 81% | 50% |
> > > | Left-only (Fig.5) | 25 | 10 | 89% | 61% | 88% | 88% | 50% |
> > > | Bidirectional (Fig.8) | 20 | 10 | 92%(93%) | 64%(68%) | 91%(92%) | 90%(91%) | 44%(50%) |
> > > | Bidirectional (Fig.8) | 25 | 10 | 82%(91%) | 56%(60%) | 82%(90%) | 82%(89%) | 40%(45%) |
> > >
> > > For the left-only setting, accuracy is measured by top-1 retrieval, where the chance level is 50% as discussed in the manuscript (Fig. 5): the model recognizes the correct object pair but cannot distinguish left from right. For the bidirectional setting, we measure whether both correct texts (e.g., "A is left of B" and "B is right of A") occupy the top-2 positions, as defined in the main text and measured for the baseline in Fig. 8. Top-1 accuracy (whether any correct text ranks first) is also shown in parentheses for reference.
> > >
> > > Across all settings, ablating the EP term alone causes the largest accuracy drop (approximately 26-28%), while ablating PP or BP alone causes only a small drop (0-2%). All three terms (EP, PP, BP) can exhibit horizontal attention gradients in the bidirectional setting, as in the left-only setting. Ablating all three together drops accuracy to at or below 50%, indicating that the attention gradient arising from positional embeddings is essential for spatial discrimination.
> > >
> > > To better understand what the ablated model can and cannot do, we analyzed the failure cases in the bidirectional setting when the EP term or all three terms (EP+PP+BP) are ablated. Among incorrect cases (where the two correct texts do not occupy the top-2), approximately 63-75% involve the semantically correct object pair but with the wrong spatial ordering in top-2. For example, when the ground truth is {"A is left of B", "B is right of A"}, the model instead places {"A is right of B", "B is left of A"} in the top-2. This suggests that the ablated model retains some ability to identify the relevant objects and select the corresponding semantic text pair, but tends to assign the wrong spatial ordering.
> > >
> > > Overall, the ablation experiments show that the attention gradient, driven primarily by the EP term with additional support from the BP and PP terms, is the causal mechanism for left-right spatial discrimination also in the bidirectional setting.
> > >
> > > We are grateful for the reviewer's detailed and insightful feedback throughout the review process. The questions on non-linearity, capacity collapse, and multi-layer analysis pushed us to develop the LayerNorm decomposition, 2D image experiments, autoregressive VLM experiments, and the ablation analysis of the bidirectional setting, which we believe significantly strengthen the paper. We will incorporate all discussed results in the revision.

---

### Official Review · Reviewer_LdK1 · 2026-03-12

**Soundness:** 2
**Presentation:** 2
**Significance:** 2
**Originality:** 2
**Overall Recommendation:** 4
**Confidence:** 4

**Summary:**

This paper shows how left–right relations understanding emerge in contrastive-based vision–language models. The authors train a small Transformer vision and text encoders with a contrastive objective on a synthetic 1D image–text dataset containing one or two objects. They evaluate the ability of the model to recognize left–right relations and generalize to unseen object pairs while varying the number of object labels and spatial layouts. The results suggest that label diversity is the main factor enabling relational generalization. The authors also provide a mechanistic analysis showing that interactions between positional and token embeddings induce an attention gradient that breaks left-right symmetry in the encoders.

**Compliance With Llm Reviewing Policy:**

Affirmed.

**Final Justification:**

The rebuttal introduces additional experiments (2D setting and autoregressive VLMs) that strengthen the paper and address my main concerns regarding generality and scope. These additions make the contribution more convincing, and I therefore increase my score to weak accept (4).

**Key Questions For Authors:**

1. Is this symmetry-breaking mechanism specific of contrastive training, or is it expected to emerge also in generative vision-language training?
2. Could this experimental setup also be used to investigate more complex spatial relations in 2D images?

**Limitations:**

Yes

**Strengths And Weaknesses:**

Strenghts:

1. The paper is clearly written and easy to follow. The figures are informative and help to understand both the experimental setup and the mechanistic analyses.
2. The attention-logit decomposition provides a mechanistic explanation for how spatial relations understanding emerge in contrastive VLM training.

Weaknesses:

1. The experiments are conducted in a very simplified 1D environment using only small models. It is unclear whether the same mechanism would appear in large CLIP models trained on natural 2D images.
2. The paper motivates its focus on contrastive CLIP-style models by stating that: “We focus on CLIP-style contrastive models because CLIP serves as the vision encoder in many downstream VLMs, so understanding its spatial encoding capabilities—or lack thereof—has broad implications.”. However, this justification is somewhat incomplete. Recent work on generative image captioning models has reported stronger compositional and relational reasoning capabilities than contrastive approaches while using a frozen CLIP vision backbone [1].
3. The analyses and experiments are restricted to the vision encoder only, assuming that it is the only responsible for the poor performance of CLIP-style models in tasks requiring spatial understanding.

[1] Parascandolo F. et al. "Causal Graphical Models for Vision-Language Compositional Understanding". ICLR 2025

---

> ### Author Rebuttal · Authors · 2026-03-31
>
> We thank the reviewer for the constructive feedback. We are glad that the reviewer finds the paper "clearly written and easy to follow" with "informative" figures, and that the attention-logit decomposition provides "a mechanistic explanation for how spatial relations understanding emerges." We address each concern and question below.
> ## W1/Q2: On extension to 2D and larger models.
>
> Encouraged by the reviewer's suggestion, ***we extended our setting from 1D to 2D.*** We place two objects in 4×4 images, assign relational texts (e.g., "A is left of B") based on their horizontal positions, and flatten the image into 16 tokens for the transformer. For simplicity, we exclude images where two objects share the same horizontal position. The model generalizes to unseen object pairs with scaling curves similar to Fig. 4(a): for example, at N_tot=20, n_2=10, single-object positional and seen-pair configuration generalization accuracy reach 100%, and unseen-pair generalization reaches 98%. ***The attention decomposition reveals the horizontal attention gradient from positional embeddings as in the 1D case.*** A natural next step is to examine whether analogous gradients emerge for above/below relations, and whether horizontal and vertical gradients can coexist to support multiple relation types simultaneously. We leave this for future work.
>
> We agree with the reviewer that validating the mechanism on models trained on natural 2D images is an important goal. While a gap remains between our toy setting and full-scale models, we believe our attention decomposition approach provides a useful analytical framework as a starting point for investigating spatial reasoning mechanisms in more realistic settings. We will discuss this perspective and include the 2D results in the revision.
>
> ## W2: On the motivation for studying CLIP.
>
> We agree that the original motivation should be strengthened, and we thank the reviewer for pointing this out. Our broader motivation is to develop mechanistic understanding of spatial reasoning in VLMs in general, not limited to CLIP. However, such mechanistic analyses remain scarce in the literature, so we started with one of the fundamental VLM architectures: CLIP. Encouraged by the reviewer's comment, we have also analyzed another fundamental architecture, an autoregressive VLM which can generalize and show attention gradient (see W3/Q1 below). We agree with the reviewer that new architectures improving compositional performance, such as Parascandolo et al. (2025), continue to be proposed, and extending mechanistic analysis to these models is important. This is, however, beyond the scope of the current paper. In the revision, we will reframe the motivation to better reflect this broader perspective.
>
> ## W3/Q1: On the scope beyond the vision encoder and contrastive training.
>
> We appreciate this perspective. Motivated by the reviewer's question, ***we trained an autoregressive VLM on the same toy setting.*** We use randomly initialized, frozen token embeddings for each object label as a minimal vision encoder, deliberately excluding positional embeddings so that spatial information must come from the decoder. The vision tokens are mapped to the text space via an MLP and fed into the decoder in spatial order. The decoder's own learnable positional embeddings provide the spatial information. We train a 2-layer attention-only Transformer decoder with next-token prediction: the input is [10 vision tokens (one per pixel)] [BOS] ["A"] ["is left of"] ["B"], and the target is ["A"] ["is left of"] ["B"] [EOT]. The vision tokens attend bidirectionally to each other, while text tokens use causal attention. The model generalizes to unseen object pairs: for example, at N_tot=40, n_2=10, it achieves 100% accuracy on single-object positional and seen-pair configuration generalization, and 94% on unseen-pair generalization, where accuracy is measured by free generation of the complete relational text. ***Notably, in the first layer of the decoder, our attention-logit decomposition revealed that text token queries exhibit a clear attention gradient over the vision token keys, preferentially attending to one side of the image sequence as observed in our CLIP case. This indicates that the symmetry-breaking mechanism is not specific to contrastive training or the vision encoder, but emerges also in autoregressive training where the decoder is responsible for spatial reasoning.*** We note that a 1-layer decoder does not generalize in this setting, though we cannot rule out that further hyperparameter tuning might resolve this. A detailed mechanistic analysis of how the attention gradient operates in the autoregressive setting, including its interaction with causal attention on text tokens, remains an important direction for future work.
>
> We thank the reviewer for encouraging us to pursue this experiment, which we believe significantly strengthens the paper. We will include the results in the revised manuscript.

---

> > ### Author Rebuttal · Reviewer_LdK1 · 2026-04-01
> >
> > I thank the authors for their thorough response, I have no further concerns and would like to raise my score.

---

> > > ### Author Response · Authors · 2026-04-02
> > >
> > > We sincerely thank the reviewer for the thoughtful feedback and for recognizing our efforts to address the concerns. The reviewer's questions, particularly on autoregressive VLMs and 2D extensions, pushed us to significantly strengthen the paper.

---

### Official Review · Reviewer_LgEH · 2026-03-13

**Soundness:** 2
**Presentation:** 2
**Significance:** 2
**Originality:** 1
**Overall Recommendation:** 3
**Confidence:** 3

**Summary:**

* This work asks whether a CLIP model can learn left-right relations and how such behavior emerges internally.
* To study this in a controlled setting, the authors use a simplified 1D synthetic image-text setup that makes the model’s behavior easier to analyze.
* The results show that the model can generalize to unseen object pairs, and argue that this is enabled by symmetry breaking driven by positional embeddings, especially through a horizontal attention gradient.
* Overall, the paper shows interpretable mechanistic explanation in a minimal setting for VLMs.

**Compliance With Llm Reviewing Policy:**

Affirmed.

**Final Justification:**

I thank the authors for the rebuttal and additional results.
The paper is clear and technically reasonable, and the rebuttal addressed some empirical concerns.
However, from a novelty perspective, I would like to maintain my original assessment.

**Key Questions For Authors:**

Please see the weaknesses above.

**Limitations:**

It would be better if discussions in more detail about how this differs from existing studies.

**Strengths And Weaknesses:**

S1.
One strength of this paper is that it provides an intuitive mechanistic interpretation through a very simplified setting.
It makes the analysis much easier to follow.

S2.
The discussion of the horizontal attention gradient is fair.
Even though the setup is admittedly artificial, the proposed mechanism is stated in a way that feels concrete rather than speculative.

S3.
The paper also benefits from having multiple pieces of supporting evidence. The main claim is not resting on a single experiment.
Within the paper’s intended scope, I found the empirical case reasonable.

W1.
One main concern is about the novelty.
Positional relations and the activations of spatially relevant tokens have already been studied in many prior work. While this paper uses an extremely simplified toy setting to make the analysis tractable, the main finding reads less like a new observation and more like a restricted case of existing discussions on positional embeddings and spatial token behavior.

* What's "up" with vision-language models? Investigating their struggle with spatial reasoning (EMNLP 2023)
* An Explainable Toolbox for Evaluating Pre-trained Vision-Language Models (EMNLP 2022)
* VALSE: A Task-Independent Benchmark for Vision and Language Models Centered on Linguistic Phenomena (ACL 2022)
* SugarCrepe: Fixing Hackable Benchmarks for Vision-Language Compositionality (NeurIPS 2025)
* GSR-BENCH: A Benchmark for Grounded Spatial Reasoning Evaluation via Multimodal LLMs (NeurIPSW 2024)
* Understanding the Limits of Vision Language Models Through the Lens of the * Binding Problem (NeurIPS 2024)
* When and why vision-language models behave like bags-of-words, and what to do about it? (ICLR 2023)

W2.
The unseen-pair result is interesting, but it still seems like a narrow instance of compositional or binding-related generalization.
Related issues (What’s Up, VALSE, Understanding the Limits of VLMs Through the Lens of the Binding Problem, When and why VLMs behave like BoWs) have already been studied in more realistic settings, especially in prior work on spatial reasoning, vision-language grounding, and the binding problem.
In that sense, the main gain here is analytical clarity rather than broader empirical coverage.

W3.
The highly simplified setting makes the analysis intuitive and easy to follow. However, I do not think the paper provides enough evidence that the same reasoning extends to (2D) CLIP models operating in realistic settings with diverse and complex spatial relations.
As a result, the explanatory value of the toy setup is clear, but its external validity remains uncertain.
It would be beneficial to see that experiments in limited settings can be extended to more general situations.

---

> ### Author Rebuttal · Authors · 2026-03-31
>
> We thank the reviewer for the careful reading. We are encouraged that the reviewer finds our mechanistic interpretation "intuitive" and "concrete rather than speculative," and the empirical case "reasonable" with the main claim "not resting on a single experiment." We address each concern below.
>
> ## W1/W2: On novelty relative to prior work
>
> We appreciate the comprehensive list of references and want to clarify how our work relates to them. The cited works include spatial relations in their evaluation benchmarks alongside attribute binding and other compositional tasks, providing valuable characterizations of where VLMs succeed and fail. Our contribution addresses both questions: we also ask whether CLIP-style contrastive training can acquire spatial reasoning, but additionally investigate through what internal mechanism this capability emerges. The cited works focus primarily on the first question, whereas our mechanistic analysis of the EP cross-term and the role of label diversity addresses the second, which remains largely unexplored. None of the cited works identify a specific attention component responsible for spatial discrimination, nor show that ablating it directly degrades performance. We view our work as complementary to the evaluative literature and will expand the related work section to make these connections more explicit.
>
> We also note that while the cited works include spatial relations in their benchmarks, attribute binding for single objects (e.g., associating "yellow" with "circle") and relational understanding between multiple objects (e.g., "A is left of B") are likely distinct problems requiring different mechanisms — the former involves associating properties with an individual object, while the latter requires comparing positions across objects. Our work contributes mechanistic understanding on the relational side, which to our knowledge has not been addressed in VLMs.
>
> We also agree with the reviewer's characterization that "the main gain here is analytical clarity rather than broader empirical coverage." We believe analytical clarity about internal mechanisms is precisely what is missing from the existing literature, and extending such mechanistic understanding to more realistic VLMs is an important goal. Our toy setting provides a necessary first step toward this, enabling the kind of targeted diagnosis we demonstrate (e.g., the role of label diversity, the minimum architectural requirements for symmetry breaking).
>
> We will include this discussion and citation in the revised manuscript.
>
> ## W3: On extension to 2D settings
>
> We take this concern seriously and ***have conducted additional experiments extending our setting from 1D to 2D.*** We place two objects in 4×4 images and assign relational text descriptions (e.g., "A is left of B") based on their horizontal positions in the 2D image. The image is then flattened into a sequence of 16 tokens for input to the transformer. The hyperparameters are the same as those used in Fig.4. For simplicity, we exclude images where the two objects share the same horizontal position. The model successfully generalizes to unseen object pairs, where the scaling curves are similar to Fig.4(a). For example, when N_tot=20, n_2=10, single-object positional, seen-pair configuration and unseen-pair generalization accuracy are 100%,100% and 98%, respectively. ***Furthermore, the attention decomposition reveals the horizontal attention gradient arising from positional embeddings as we identified in the 1D case.*** This demonstrates that the mechanism is not an artifact of the 1D simplification. We agree with the reviewer that a gap remains to full-scale CLIP models operating on natural 2D images with diverse spatial relations. We regard this as an important long-term goal.
>
> Furthermore, following the reviewer's suggestion to extend to more general situations, ***we also tested whether the mechanism generalizes beyond the contrastive objective. We trained an autoregressive VLM on the same toy setting with a 2-layer attention-only Transformer decoder and next-token prediction (see reply to W3/Q1 by Reviewer LdK1). The model generalizes to unseen object pairs (e.g. 94% unseen-pair generalization accuracy), and the attention gradient appears in the vision tokens of the decoder.*** This demonstrates that the symmetry-breaking mechanism extends beyond CLIP-style contrastive training to a fundamentally different learning paradigm.
>
> We will include these results in the revised manuscript.

---

> > ### Author Rebuttal · Reviewer_LgEH · 2026-04-05
> >
> > Thank you for the thoughtful rebuttal and additional clarifications.
> > I have carefully read the response and appreciate the effort to address the reviewers’ comments.
> > While the rebuttal strengthens several aspects of the paper, my overall assessment remains unchanged in terms of novelty and scope.
> > I will maintain my original score.

---

> > > ### Author Response · Authors · 2026-04-05
> > >
> > > We thank the reviewer for the thoughtful engagement with our work and for acknowledging that the concerns have been "fully resolved."
> > >
> > > We would also like to briefly highlight what we believe is novel in our contribution: to our knowledge, this is the first work to identify a specific mechanistic pathway (attention gradient driven by positional embedding) responsible for spatial relation learning in vision-language models, and to provide causal evidence through ablation that this mechanism drives spatial discrimination in our controlled setting. During the rebuttal, we further demonstrated that this mechanism generalizes to 2D images and persists in an autoregressive VLM architecture. We believe these findings complement existing evaluative studies of spatial reasoning in VLMs by providing mechanistic understanding of the underlying mechanism, which has been largely unexplored. This approach follows a productive tradition in mechanistic interpretability, where controlled minimal settings have yielded insights that later generalized to larger models (e.g., Elhage et al., 2021; Olsson et al., 2022). We recognize that extending the analysis to natural images and large-scale VLM models is an important future direction, and we look forward to exploring this.
> > >
> > > Elhage et al., 2021: _A Mathematical Framework for Transformer Circuits._ Transformer Circuits Thread (Anthropic), 2021.
> > > Olsson et al., 2022: _In-context Learning and Induction Heads._ Transformer Circuits Thread (Anthropic), 2022.

---

### Official Review · Reviewer_REMU · 2026-03-13

**Soundness:** 2
**Presentation:** 4
**Significance:** 2
**Originality:** 2
**Overall Recommendation:** 4
**Confidence:** 4

**Summary:**

The paper studies how CLIP-style VLMs learn left-right spatial relations in a simplified synthetic 1D setting. The authors train transformer-based "vision" and text encoders on one- and two-object scenes and evaluate generalization to unseen object pairs under varying levels of label diversity and layout diversity. They find that increasing label diversity improves generalization more than increasing layout diversity. To explain this, they perform a mechanistic analysis of attention in a simplified model and argue that interactions between token and positional embeddings break left-right symmetry and enable the model to perform well.

**Compliance With Llm Reviewing Policy:**

Affirmed.

**Final Justification:**

The conducted experiments in the rebittal with three objects strengthens the contribution a lot and helps prevent the considered models of cheating by requiring them to model relative positions between object. The results claimed in the paper appear to extend to three object setting and that's a useful result.

I hope the authors can highlight these results and conduct additional experiments under 3+ objects to stress test their ideas further.

**Key Questions For Authors:**

See above.

**Limitations:**

Yes.

**Strengths And Weaknesses:**

Strengths:
1. The paper studies a common failure case in CLIP models: spatial understanding. I think the problem is very interesting and the study is well motivated.
2.  The approach to understanding the problem is very clean and fitting. I think the controlled approach allows isolating the main reasons for failure cases.
3. The paper is well written and easy to follow.



Weaknesses

1. The main empirical finding — that label diversity matters more than layout diversity — is interesting, but it could use some context. For example [1] discusses data diversity and its impact on generalization in a related setting. I think discussing the impact of data diversity and how it relates to prior work would be useful.

[1] Uselis, Arnas, Andrea Dittadi, and Seong Joon Oh. "Does Data Scaling Lead to Visual Compositional Generalization?." arXiv preprint arXiv:2507.07102 (2025).

2. My main concern is that the task may not impose any special structure on the embedding space and may not express relational understanding in the first place. The reason for this is that with only two objects, it seems sufficient for the model to learn role-conditioned object codes and compose them additively, i.e. something like $f(x,y) = u_{x,\text{left}} + u_{y,\text{right}}$. That is, each unique object could be represented as either `left` or `right`. If such a structure is indeed possible, then the unseen-pair generalization is not evidence of a mechanism CLIP may employ in spatial understanding -- it would mean that the studied problem is too simple. And importantly, under this interpretation, label diversity helping more than layout diversity is exactly what you'd expect from prior work on data diversity and linear factorization [1]: more objects means more role-conditioned components to recombine, and that is all that is needed. The current experiments do not rule this out, and I think they should. The reason I bring this structure up in particular is because [1] showed that such a structure tends to arise in models trained from scratch which matches this paper's setting, though [1] setting is simpler.

3. The only way forward, as far as I can tell, is scaling to 3+ objects in the experiments (ideally up to 10). With three or more objects, the model would need to represent multiple simultaneous relations and be consistent. This would also approximate realistic conditions of multiple scenes where an abundance of relationships can be derived. I expect generalization to become almost impossible at that scale, but that's expected and would still result in a strong paper. Otherwise, a convincing argument that the shortcut mentioned above is not possible would also address this concern.

4. In general, I found the empirical results to be rather small in scale. Assuming the concerns above are addressed, I think the submission would benefit from larger scaling settings in terms of number of objects and layout sizes.


Nit:
The connection to grokking seems speculative (L202). As written it is not developed enough to be convincing

Showing that the results generlize to 3+ objects would lead to an increase in score.

---

> ### Author Rebuttal · Authors · 2026-03-31
>
> We thank the reviewer for the thoughtful and constructive feedback. We are encouraged that the reviewer finds the problem "very interesting and well motivated," the approach "very clean and fitting," and the presentation "excellent." We address the remaining concerns below.
>
> ## W1/W2: On the relationship to data diversity and Uselis et al. [1].
>
> We thank the reviewer for this reference. Both papers observe that data diversity drives generalization, but the types of generalization are not identical. In [1], the task is compositional generalization of context-free attributes: "red" means "red" regardless of the accompanying shape, and diversity forces an additive factorization (r = u + v). In our setting, the task is relational generalization: "left" is not a property of an object but a comparison between two positions. The same object A at position x=3 is "left of B" or "right of B" depending entirely on B's position.
>
> Our attention decomposition reveals how the model handles this: positional-token interactions (the EP cross-term) induce a position-dependent attention gradient providing directional selectivity that applies regardless of which objects occupy those positions. Crucially, ablating the positional embedding contributions from the attention terms (Fig. 5(e)) eliminates the model's ability to discriminate spatial relations, confirming that the mechanism we identified is necessary for spatial understanding. Label diversity forces this mechanism to generalize across object identities, rather than relying on object-specific shortcuts. This differs from [1], where diversity forces independent directions for each concept value. We will include this discussion in the revision.
>
> ## W3: Experiments with 3+ object images.
>
> We agree that extending to multi-object images is important. Encouraged by this suggestion, ***we extended evaluation to 3-object scenes, where an object can be simultaneously right of one neighbor and left of another, making a fixed two-role assignment insufficient.***
>
> Interestingly, models trained on single- and 2-object images (i.e., the models from Fig. 4) show partial transfer to 3-object spatial discrimination. We evaluate on 3-object scenes composed of object labels that do not appear in 2-object training dataset (i.e., OOD generalization to unseen pairs), where three pairwise relations hold simultaneously (A left of B, A left of C, B left of C). The best performing model retrieves at least one correct relation as its top-1 prediction in approximately 83% of cases, substantially above chance. However, it recovers on average about 1.8 out of 3 correct relations in its top-3 predictions, and retrieves all three simultaneously in at most 15% of cases. These results suggest that the spatial mechanism transfers to individual pairwise judgments but does not fully compose to handle multiple relations at once.
>
> We believe this partial transfer is informative. The attention gradient mechanism should in principle support pairwise spatial discrimination even with three objects, since it operates over positions regardless of how many objects are present. Indeed, the 83% top-1 accuracy confirms this. The difficulty lies not in the spatial mechanism itself, but in the representation bottleneck: the single CLS embedding must aggregate information from all three objects simultaneously, and when matched against a text describing only two objects, the third object acts as noise. We also attempted joint training with 3-object images, but this did not improve, likely due to the contrastive objective treating valid but unselected text descriptions as negatives. Addressing this through modified training objectives requires further investigation. These results suggest that the core limitation is in how contrastive models compress multi-object scenes into a single vector, rather than in the spatial reasoning mechanism itself. We will include these results in the revised manuscript.
>
> ## W4: On the scale of experiments.
>
> We acknowledge the importance of larger-scale validation as a key future direction. Within the current study, ***we have extended experiments along three axes: (1) 2D toy images (4×4), where the same attention gradient emerges (see reply to Reviewers LgEH and LdK1); (2) 3-object scenes as above; and (3) an autoregressive VLM trained with next-token prediction, where the model generalizes (e.g. 94% unseen-pair generalization accuracy) and the attention gradient appears in the vision tokens (see reply to Reviewer LdK1).*** We find these results encouraging as a basis for future investigations at larger scale. We will include these results in the revised manuscript.
>
> ## On the connection to grokking (L202).
>
> We agree this reads as stronger than intended. We will soften the language and clarify that we do not claim a shared mechanism.

---

> > ### Author Rebuttal · Reviewer_REMU · 2026-04-03
> >
> > I don't think the paper warrants acceptance unless it extends their setting to 3+ objects and verifies the mechanism they hypothesize is found there. Otherwise, the trained models are susceptible to shortcuts and therefore aren't reflecting real-world complexities

---

> > > ### Author Response · Authors · 2026-04-07
> > >
> > > We thank the reviewer for the follow-up. Encouraged by the reviewer's suggestion, we have continued experiments throughout the rebuttal period. ***We have now trained models that generalize to 3-object images, and found that the attention gradient mechanism is present in this setting.***
> > >
> > > ## Training with 3-object images
> > >
> > > We trained models on single-, 2-, and 3-object images jointly via contrastive learning. For 3-object images of the training dataset, we used only the labels that appear in the 2-object training dataset. Since each 3-object image has three valid text descriptions, we randomly select one as the text pair in the training dataset. We evaluate on 3-object images composed of object labels that do not appear in any 2-object or 3-object training data (i.e., OOD generalization to unseen labels) using the following metrics: "3obj top-1" (whether any correct text ranks first), "3obj top-3 mean" (average number of correct texts in the top-3), and "3obj top-3 all" (whether all three correct texts appear in the top-3). "2obj OOD" is the unseen-pair generalization accuracy for 2-object images.
> > >
> > > All models hereafter use attention-only encoders (without MLP or LayerNorm). With a single-layer vision encoder, the models achieve 35% (trained with 1-layer text encoder) and 56% (trained with 2-layer text encoder) on "3obj top-3 all", both significantly above chance, demonstrating that generalization to 3-object scenes emerges (N_tot=20). We note that these results are preliminary and further parameter tuning may improve performance for both configurations. With 2-layer vision and 2-layer text encoders, performance improves further. For example, a model with strong generalization achieves 98% on "3obj top-1" and 82% on "3obj top-3 all" (N_tot=20), as summarized in the table below.
> > >
> > >  | Ablation | 2obj OOD | 3obj top-1 | 3obj top-3 mean | 3obj top-3 all |
> > >  |---|---|---|---|---|
> > >  | Baseline w/o abl. | 95% | 98% | 2.82 | 82% |
> > >  | EP abl. | 65% | 69% | 1.98 | 28% |
> > >  | PP abl. | 95% | 97% | 2.81 | 81% |
> > >  | BP abl. | 95% | 98% | 2.81 | 82% |
> > >  | EP+PP+BP abl. | 52% | 58% | 1.68 | 17% |
> > >
> > > In both single-layer and 2-layer vision encoder cases, we found that horizontal attention gradients emerge in the EP, PP, and BP terms in the first layer of the vision encoder. As an example from the 2-layer vision encoder model (see table above), ablating the EP term alone causes the largest drop (e.g., 82% → 28% on “3obj top-3 all”), while ablating PP or BP alone has minimal effect. Ablating all three gradient-bearing terms together drops accuracy further (e.g., 82% → 17%). These results indicate that the attention gradient mechanism, driven primarily by the EP term, is causally relevant in the 3-object setting. In the 2-layer model, the attention gradient is present and causally important in the first layer, but other mechanisms in the second layer may also contribute to the improved performance. Disentangling the contributions of each layer and exploring whether 1-layer models can achieve comparable performance with alternative training strategies are important directions for future work. We will include the results in the revision.
> > >
> > > We believe these results directly address the reviewer's concern: the model is trained on 3+ objects, generalizes to unseen triplets, and the attention gradient mechanism is verified through both observation and ablation. We thank the reviewer for pushing us to conduct this important experiment, which we believe significantly strengthens the paper.

---

### Decision · Program_Chairs · 2026-04-30

**Decision:**

Accept (regular)

**Comment:**

The paper studies whether/how CLIP-style vision language models can learn to distinguish left-right relations in a synthetic 1D dataset, find increasing label diversity to be more important than  layout diversity for generalization, and also provide a mechanistic explanation through attention decomposition. The reviewers acknowledged the relevance of the studied problem, clear motivation and presentation, intuitive constructed dataset and mechanistic explanation. The concerns that were consistently expressed by reviewers were missing discussion and novelty w.r.t. prior work, limited scope to synthetic scenes, simplified spatial relations only, unclear validity of the claims in more complex architectures and real-world benchmarks. Given that this is an important and interesting problem along with the limited scope of the setup, the AC suggests weak accept.